# Pruning-Robust Mamba with Asymmetric Multi-Scale Scanning Paths

**Jindi Lv**[1]   **Yuhao Zhou**[1][§]   **Mingjia Shi**[2]   **Zhiyuan Liang**[3]   **Panpan Zhang**[3]
**Xiaojiang Peng**[3]   **Wangbo Zhao**[3]   **Zheng Zhu**[4]   **Jiancheng Lv**[1]   **Qing Ye**[1][†]   **Kai Wang**[3]
[1]Sichuan University   [2]University of Virginia   [3]National University of Singapore   [4]GigaAI

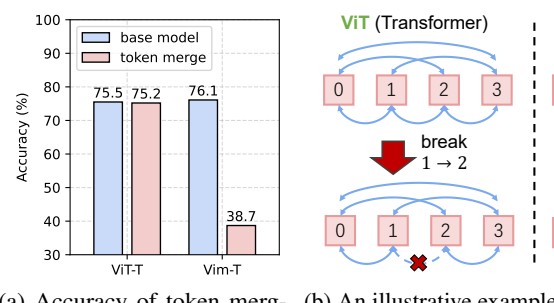

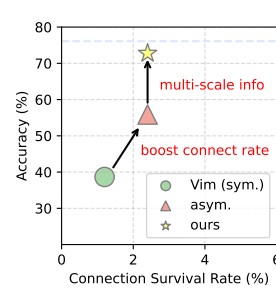

(a) Accuracy of token merging: ViT-T (Transformer) vs. Vim-T (SSM).

(b) An illustrative example showing the impact of token connection disruption in ViT (Transformer) versus Vim (SSM).

(c) Correlation between token connection survival rate and accuracy.

Figure 1: Comparative analysis of token reduction. In Figure (b), blue solid lines denote direct token connections, whereas black dotted lines signify potential connections.

## Abstract

Mamba has proven efficient for long-sequence modeling in vision tasks. However, when token reduction techniques are applied to improve efficiency, Mamba-based models exhibit drastic performance degradation compared to Vision Transformers (ViTs). This decline is potentially attributed to Mamba's chain-like scanning mechanism, which we hypothesize not only induces cascading losses in token connectivity but also limits the diversity of spatial receptive fields. In this paper, we propose Asymmetric Multi-scale Vision Mamba (AMVim), a novel architecture designed to enhance pruning robustness. AMVim employs a dual-path structure, integrating a window-aware scanning mechanism into one path while retaining sequential scanning in the other. This asymmetry design promotes token connection diversity and enables multi-scale information flow, reinforcing spatial awareness. Empirical results demonstrate that AMVim achieves state-of-the-art pruning robustness. During token reduction, AMVim-T achieves a substantial 34% improvement in training-free accuracy with identical model sizes and FLOPs. Meanwhile, AMVim-S exhibits only a 1.5% accuracy drop, performing comparably to ViT. Notably, AMVim also delivers superior performance during pruning-free settings, further validating its architectural advantages.

## 1   Introduction

In recent years, the Mamba architecture, built upon state space models (SSMs), has emerged as a transformative paradigm for efficiently modeling long-range dependencies in vision tasks [1, 2, 3, 4, 5]. By introducing an innovative chain-like scanning mechanism, Mamba [6, 7, 8, 9, 10, 11] successfully reduces the computational complexity from the quadratic demands of Transformers [12, 13, 14] to a linear scale. Alongside these advancements, token reduction techniques (eg., pruning [15, 16, 17]

39th Conference on Neural Information Processing Systems (NeurIPS 2025).

and merging [18, 19]) for Mamba have garnered increasing attention as promising avenues toward further optimization.

Nevertheless, Mamba exhibits significantly greater performance degradation during token reduction compared to Transformers (eg., ViT [13]), as shown in Figure 1a. This discrepancy arises from Transformers using self-attention to establish fully connected token relationships, while Mamba processes tokens sequentially along chain-like scanning paths. This chain-based structure makes Mamba susceptible to cascading information loss during token reduction. For clarity, an illustrative example is provided in Figure 1b.

Recent Mamba variants [20, 21, 22], such as Vim [6], have attempted to mitigate this limitation through **dual-path** scanning strategies that combine forward and reverse sequential paths. While these symmetric designs enhance sequence modeling capabilities and improve baseline performance, they remain ineffective for token reduction. The inherent symmetry of dual-path scanning confines token relationships to the same chain-like structure, failing to address the systemic vulnerability to large-scale connection disruption during pruning.

Inspired by these observations, we hypothesize that minimizing connection disruption during token reduction can mitigate performance drop. To validate this, we introduce asymmetric scanning into dual-path Mamba. As illustrated in Figure 1c, asymmetric scanning paths reduce accuracy degradation from 38% (with symmetric paths) to 21% at the same pruning ratio. This suggests that diversifying chain-like dependencies effectively mitigate pruning-induced performance decline.

To further elucidate this phenomenon, we quantify the token connection survival rate across different dual-path strategies during token reduction. Figure 1c reveals a strong positive correlation between connection survival rates and accuracy, with asymmetric paths exhibiting superior robustness. This confirms our hypothesis: enhancing token connection survival rates via asymmetric path diversification is pivotal for improving Mamba's pruning resilience.

In this work, we propose **AMVim**, a novel **A**symmetric **M**ulti-scale **Vi**sion **M**amba for pruning robustness. To enhance space information diversity, we integrate a window-aware scanning mechanism into one path. By adopting a different scanning direction within windows compared to the main path, we construct multi-level asymmetric paths. This multi-dimensional information flow enables each token to perceive neighborhood information from multiple perspectives. Furthermore, the integration of window-based scanning with the main path creates a multi-level complementary design, allowing for interactions between global context and local dependencies.

Empirically, as shown in Figure 1c, our method results in just a 3% accuracy drop during token reduction (with the blue dotted line representing the baseline accuracy of 76.1%) on ImageNet-1K, achieving a 34% improvement over Vim. This highlights that the multi-scale scanning mechanism enhances the spatial awareness of SSMs, significantly reducing token sensitivity to local variations.

We highlight the main contributions of this paper below:

- We hypothesize token connection survival rate is a critical factor in performance degradation and propose asymmetric scanning paths to effectively mitigate this issue.
- We design a multi-scale asymmetric scanning mechanism that balances global and local spatial information while preserving the benefits of asymmetric paths.
- Our method achieves state-of-the-art pruning resilience, outperforming Vim-T by 34% on ImageNet-1K under identical parameters and FLOPs.

## 2 Related Work

### 2.1 State Space Models

SSMs [23, 24, 25, 26, 24, 27] were initially proposed in the NLP community to model long-range dependencies in text. Recently, SSM variants have emerged as effective alternatives to ViTs [13, 14, 12, 28, 29, 30], reducing computational complexity in visual tasks from quadratic to linear time. S4ND [31] is the first work to apply SSMs to visual tasks, extended the S4 [32] model by normalizing the parameters to a diagonal structure. However, this approach struggled to capture image information in an input-dependent manner. In response, Vim [6] was proposed, introducing bidirectional scanning to enhance spatial awareness in vision tasks. Building upon this, PlainMamba [33] introduced

continuous 2D scanning, which improves spatial continuity by maintaining adjacency among tokens within the scanning sequence. Moreover, VMamba [7] proposed an SS2D scanning mechanism that enables comprehensive scanning across four distinct paths.

Despite significant advancements, recent studies [34, 35] have identified limitations in Mamba's chain-like scanning structure, particularly in capturing local spatial dependencies. To address this, Shi et al. [35] introduced a multi-scale 2D scanning technique based on VMamba, combining original and downsampled feature maps to alleviate the long-range forgetting issue. Similarly, LocalMamba[34] proposed a window-based scanning mechanism that dynamically selects search paths at each layer to capture local dependencies. In contrast, we introduce a multi-scale asymmetric scanning mechanism. By improving both direct token connection complementarity and multi-scale information synergy, our method enhances the spatial perception capability of SSM.

## 2.2 Token Reduction

Token reduction aims to enhance computational efficiency by dynamically removing or consolidating redundant tokens during inference. These methods are typically classified into token pruning [15, 19, 17, 16] and token merging [18, 36, 37, 38, 39]. Token pruning identifies and eliminates low-importance tokens. For example, DynamicViT [15] employs the Gumbel-Softmax strategy to prune less informative tokens, while EViT [19] relies on the attentiveness of the [CLS] token to determine key tokens. In contrast, token merging combines semantically similar tokens, as demonstrated by ToMe [18], a training-free approach that merges tokens via bipartite matching. However, the underlying architectural differences between ViTs and Mambas present unique challenges when applying these techniques to Mambas. First, most token pruning methods are designed for ViTs and rely on self-attention scores, which Mamba lacks, making direct transfer infeasible. Second, token merging in Mamba leads to significant performance degradation due to its chain-like scanning mechanism, which enforces rigid sequential dependencies between tokens.

Recent work has sought to address these challenges. For instance, Zhan et al. [40] proposed a pruning-aware hidden state alignment method to selectively skip tokens in Mamba. Their follow-up work [41] introduced hybrid metrics combining token importance and similarity for pruning. These approaches focus on designing specialized pruning strategies for Mamba. In contrast, our approach aims to strengthen the intrinsic robustness of Mamba's architecture for token reduction. By mitigating chain dependency vulnerabilities, our method enables seamless integration of existing token merging techniques while further improving performance, offering a significant advantage for practical deployment.

# 3 Method

## 3.1 Preliminaries

SSMs map a 1D input sequence $x(t) \in \mathbb{R}$ to an output sequence $y(t) \in \mathbb{R}$ through an implicit latent state $h(t) \in \mathbb{R}^N$, governed by linear ordinary differential equations (ODEs):

$$h'(t) = \mathbf{A}h(t) + \mathbf{B}x(t), \quad y(t) = \mathbf{C}h(t), \tag{1}$$

where $\mathbf{A} \in \mathbb{R}^{N \times N}$ governs state transitions, $\mathbf{B} \in \mathbb{R}^{N \times 1}$ projects inputs to the state space, and $\mathbf{C} \in \mathbb{R}^{1 \times N}$ maps the state to outputs. A defining feature of SSMs is their chain-like scanning path, which processes inputs sequentially to achieve linear computational complexity. This sequential dependency can be formalized in the discretized recurrence:

$$h_t = \overline{\mathbf{A}}h_{t-1} + \overline{\mathbf{B}}x_t, \quad y_t = \mathbf{C}h_t. \tag{2}$$

where $\overline{\mathbf{A}} = e^{\Delta \mathbf{A}}$ and $\overline{\mathbf{B}} = (\Delta \mathbf{A})^{-1}(e^{\Delta \mathbf{A}} - \mathbf{I}) \cdot \Delta \mathbf{B}$ are derived from the continuous-time parameters via zero-order hold (ZOH) discretization. The chain-like structure ensures that each token $x(t)$ interacts directly with its immediate predecessor $x(t-1)$, propagating information sequentially through the state $h(t)$.

Mamba enhances SSMs with input-dependent selectivity (S6), dynamically adjusting parameters $\mathbf{B}, \mathbf{C}, \Delta$ based on $x(t)$:

$$\mathbf{B}_t = \text{Linear}_B(x_t), \quad \mathbf{C}_t = \text{Linear}_C(x_t), \quad \Delta_t = \text{Softplus}(\text{Linear}_\Delta(x_t)). \tag{3}$$

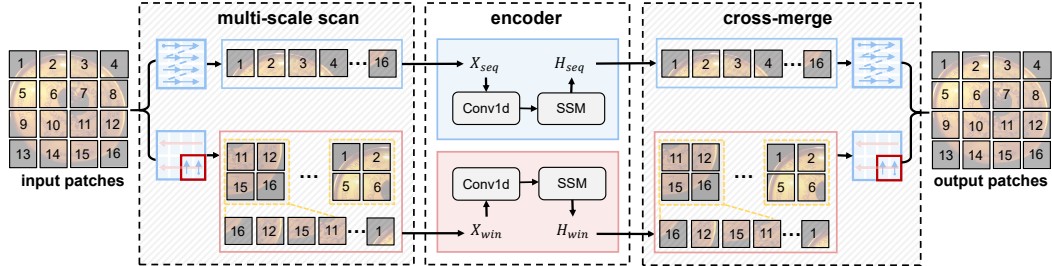

Figure 3: Illustration of the Window-based Multi-Scale State Space (WMS3) block. Input patches are processed along two scanning paths at different scales (multi-scale scan), with each sequence independently encoded (encoder). The outputs are then restored to sequential order and merged to form a 2D feature map as the final result (cross-merge).

While this improves flexibility, the underlying chain-like scanning path remains a core component. Importantly, this chain structure design introduces a critical limitation: the model relies on local neighbor dependencies for information propagation, which amplifies sensitivity to token removal.

## 3.2 Multi-Scale 2D Selective Scanning

The sequential scanning operation in S6 works well for time-series data in NLP tasks but struggles with non-causal visual data, which is inherently non-sequential and spatially complex. To address this, Vim [6] introduces bidirectional symmetric scanning paths (Figure 2a) to enhance spatial context. Although this design improves spatial context modeling, it fails to resolve the performance collapse during token reduction.

We attribute this limitation to the chain-like scanning mechanism underlying Mamba, which enforces a rigid neighbor dependency as depicted in Equation 2. To mitigate this issue, we propose a simple yet effective solution: asymmetric scanning paths (Figure 2b). By diversifying the scanning directions, asymmetric paths enhance the complementarity of token connections, as shown in Figure

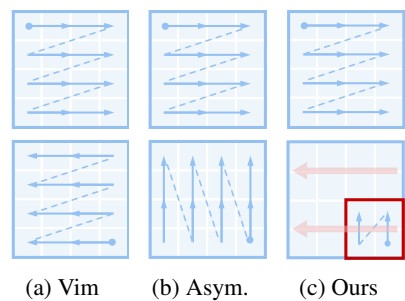

(a) Vim    (b) Asym.    (c) Ours

Figure 2: Comparison of dual-path scanning strategies. The solid dot represents the starting position.

1c, leading to a significant boost in pruning robustness. However, achieving full token connection complementarity through scanning paths alone remains challenging under computational constraints.

Recent studies [34, 35] emphasize the critical role of local spatial awareness in enhancing Mamba's robustness. Inspired by this insight, we propose a window-based multi-scale selective scanning mechanism. As shown in Figure 2c, this design integrates a window-aware scanning strategy into one path while preserving the global scanning directions of Vim. Specifically, the tokens are partitioned into non-overlapping windows, where tokens are scanned vertically within windows and horizontally globally. This orthogonal design ensures asymmetric dependencies, minimizing redundant token connections while maximizing connection complementarity. Moreover, by harmonizing global structures and local textures, it enables more comprehensive spatial understanding.

## 3.3 Overall Model Architecture

In this work, we extend Vim [6] by introducing the Window-based Multi-Scale State Space (WMS3) block. As shown in Figure 3, WMS3 operates through three sequential stages: multi-scale scanning, encoding, and cross-merge. Given an input token sequence $\mathbf{X} \in \mathbb{R}^{L \times D}$, where $L$ is the sequence length and $D$ is the feature dimension, WMS3 first reorders $\mathbf{X}$ along two distinct traversal paths: the sequential path and the window-aware path.

Each reordered sequence is then independently processed by a dedicated encoder block, which integrates a 1D convolutional layer and a S6 module. Formally, for a sequence $\mathbf{X}_p$ along path $p \in$

{sequential, window}, the encoder block computes:

$$\mathbf{H}_p = \mathrm{E}(\mathbf{X}_p) = \mathrm{S6}(\mathrm{Conv1D}(\mathbf{X}_p)) \tag{4}$$

where $\mathrm{Conv1D}(\cdot)$ enhances local feature interactions, and $\mathrm{S6}(\cdot)$ models long-range dependencies via selective state transitions.

Finally, the outputs from both paths are restored to their original spatial order and merged via a cross-merge operation:

$$\mathbf{H} = \mathrm{Linear}(\mathbf{H}_{\text{sequential}} + \mathbf{H}_{\text{window}}),$$

where $\mathrm{Linear}(\cdot)$ denotes a linear projection layer. Beyond the WMS3 block, our architecture retains the core design of Vim [6], including the placement of the [CLS] token at the sequence center. This multi-scale design expands spatial perception granularity without additional computational overhead, significantly improving robustness to token reduction while maintaining efficiency.

## 4 Experiment

### 4.1 Datasets and Settings

We evaluate AMVim on the ImageNet-1K dataset, which includes 1,000 object classes, 1.28 million training images, and 50,000 validation images. Images are augmented and resized to 224×224 for evaluation. This study focuses on the ImageNet-1K classification task, and we report top-1 validation accuracy. All experiments are conducted with 4 × NVIDIA L40S GPUs.

The window size of AMVim is set to 3×3. AMVim is fine-tuned for 150 epochs with AdamW optimization, initialized using the publicly available weights of Vim [6]. A batch size of 128 is used with two-step gradient accumulation, resulting in an effective total batch size of 1,024. Additional training details are listed in Table 9 in Appendix.

During token reduction, we employ the ToMe technique [18] by default. To ensure a fair comparison, token merging is applied to the even-indexed blocks, covering a total of 12 layers. In each layer, [5, 8, 11, 14] tokens are pruned, corresponding to reduction ratios of [0.17, 0.27, 0.36, 0.46].

### 4.2 Pruning Robustness Analysis

**Comparison with SOTA methods.**

To validate the advancement of the proposed architecture, we compare AMVim with two state-of-the-art token pruning techniques specifically designed for Mamba: Token Recognition [19] and Hidden State Alignment [40]. For a fair comparison, these two methods are evaluated on Vim [6] with a dual-path structure. Additionally, we adhered to their fine-tuning protocols [40] during token reduction and reported the final fine-tuned accuracy.

The results, presented in Table 1, show that AMVim consistently achieves the highest accuracy across various model scales while maintaining comparable FLOPs. Specifically, AMVim-T outperforms Token Recognition by 3.7% in accuracy, while AMVim-S achieves a 4.7% improvement. Furthermore, AMVim-S exhibits only a 1% accuracy drop while reducing FLOPs by one-quarter, significantly surpassing both Token Recognition and Hidden State Alignment. These results highlight that instead of designing specialized token pruning methods for Mamba, addressing its architectural fragility offers a more promising solution.

Table 1: Performance comparison with token pruning methods designed for Mamba on ImageNet-1K classification. AMVim achieves the highest top-1 accuracy across different model scales while maintaining comparable FLOPs.

| method | reduction ratio | top-1 acc.(%) | | FLOPs (G) | |
|---|---|---|---|---|---|
| | | Tiny | Small | Tiny | Small |
| Vim (baseline) | 0.00 | 76.1 | 80.5 | 1.45 | 5.08 |
| Token Recognition [19] | 0.17 | 71.3 | 74.8 | 1.28 | 3.57 |
| Hidden State Alignment [40] | 0.17 | 75.1 | 78.8 | 1.29 | 3.60 |
| AMVim-ToMe (finetune) | 0.17 | **75.3** | **79.5** | **1.27** | 3.60 |

**Robustness across various reduction ratios.** Figure 4a compares the top-1 accuracy of our method with recent state-of-the-art vision Mamba models across various token reduction ratios. As the reduction ratio increases, all methods exhibit a predictable decline in performance due to increased information loss. Our method consistently demonstrates superior robustness, outperforming Vim [6] by approximately 30% across all reduction ratios. LocalVim [34] shows improved robustness over

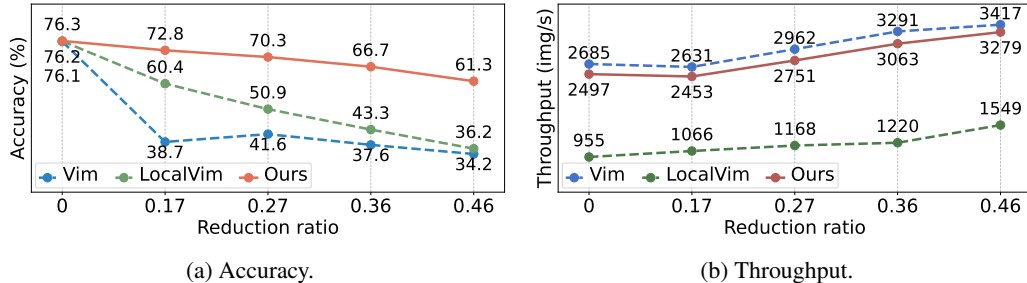

(a) Accuracy.             (b) Throughput.

Figure 4: Comparison with state-of-the-art vision Mamba methods in terms of top-1 accuracy and throughput across various token reduction ratios. Our method demonstrates superior robustness against Vim and LocalVim, while achieving comparable throughput with Vim and twice that of LocalVim. This highlights that our method strikes a good balance between efficiency and robustness.

Vim by incorporating local modeling. However, it still lags significantly behind our method, especially as the reduction ratio increases. Notably, even at a nearly threefold higher reduction ratio (0.46 vs. 0.17), our method still achieves higher accuracy than both Vim and LocalVim.

Figure 4b presents the computational throughput across methods. Our approach achieves throughput comparable to Vim, with a marginal deficit of 200 img/s attributable to directional change operations. In contrast, it delivers approximately 2× higher throughput than LocalVim, which incurs heavier computational overhead from per-layer directional changes. These results collectively underscore our method's optimal trade-off between performance robustness and computational efficiency.

**Robustness on various pruning methods.**

To further validate the pruning robustness of AMVim, we introduce random token pruning, a method that randomly discards tokens without relying on any pruning criteria (e.g., similarity or importance). Table 2 compares the top-1 accuracy of Vim and AMVim under both token merging and random pruning conditions. The results demonstrate that AMVim consistently outperforms Vim by approximately 30% across all scenarios, irrespective of the pruning method employed. This significant performance gap highlights the exceptional robustness of AMVim's architecture, which consistently delivers strong performance under any pruning method, solidifying its design superiority over Vim.

Notably, AMVim exhibits inferior performance under random token pruning compared to token merging. This observation is expected, as token merging leverages similarity as a guiding metric, whereas random pruning lacks such guidance. In contrast, at reduction rates of 0.17 and 0.27, random pruning yields better performance than token merging on Vim. This counterintuitive result suggests

Table 2: Random pruning vs. Token merging: top-1 accuracy (%) under various token reduction techniques. △ denotes the performance difference between Vim and AMVim. AMVim achieves roughly 30% higher accuracy than Vim across both token merging and token pruning methods.

| operation | reduction ratio | Vim-T (%) | AMVim-T (%) | △ (%) |
|---|---|---|---|---|
| pruning | 0.17 | 42.7 | 72.4 | 29.7↑ |
| | 0.27 | 45.9 | 69.5 | 23.6↑ |
| | 0.36 | 36.6 | 65.2 | 28.6↑ |
| | 0.46 | 22.5 | 58.6 | 32.1↑ |
| merging | 0.17 | 38.7 | 72.8 | 34.1↑ |
| | 0.27 | 41.6 | 70.3 | 28.7↑ |
| | 0.36 | 37.6 | 66.7 | 29.1↑ |
| | 0.46 | 34.2 | 61.3 | 27.1↑ |

Table 3: Performance comparison between Vim and AMVim on ImageNet-1K. "ToMe" indicates token reduction applied via ToMe [18] for each method, with training-free accuracy reported. △ represents the performance gap between Vim and AMVim. AMVim consistently outperforms Vim across both pruned and non-pruned settings.

| method | image size | #param (M) | FLOPs (G) | top-1 acc. (%) | △ (%) |
|---|---|---|---|---|---|
| Vim-T | $224^2$ | 7 | 1.5 | 76.1 | 0 |
| Vim-S | $224^2$ | 26 | 5.1 | 80.5 | 0 |
| ToMe-Vim-T | $224^2$ | 7 | 1.3 | 38.7 | 0 |
| ToMe-Vim-S | $224^2$ | 26 | 4.4 | 78.4 | 0 |
| AMVim-T | $224^2$ | 7 | 1.5 | **76.3** | 0.2↑ |
| AMVim-S | $224^2$ | 26 | 5.1 | **80.7** | 0.2↑ |
| ToMe-AMVim-T | $224^2$ | 7 | 1.3 | **72.8** | 34.1↑ |
| ToMe-AMVim-S | $224^2$ | 26 | 4.4 | **79.2** | 0.8↑ |

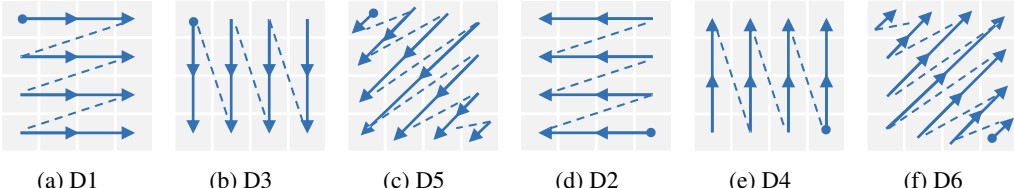

| (a) D1 | (b) D3 | (c) D5 | (d) D2 | (e) D4 | (f) D6 |

Figure 5: Illustration of the scanning directions commonly used in vision Mamba. The solid dot represents the starting position.

that Vim's performance degradation is not inherently tied to the pruning method but rather stems from its architectural vulnerability, rendering the guiding metric ineffective. These findings collectively highlight the resilience of AMVim's architecture while exposing fundamental limitations in Vim's.

### 4.3 Image Classification

Table 3 compares the performance of Vim and AMVim on the ImageNet-1K classification task with an image size of 224×224. The results show that AMVim consistently achieves higher top-1 accuracy than Vim across various model scales while maintaining the same parameter count and FLOPs. This improvement can be attributed to AMVim's enhanced spatial awareness capability, which effectively achieves the learning of both local dependency and global context.

With token reduction applied (reduction ratio = 0.16), AMVim-T exhibits a training-free accuracy drop of only 3.8%, while AMVim-S shows an even smaller drop of 1.5%, approaching the performance of ViT. Compared to Vim, AMVim-T delivers a substantial 34.1% improvement in pruned accuracy, and AMVim-S shows a 0.8% increase. These results collectively demonstrate that AMVim not only delivers superior robustness to token reduction but also enhances model expressive capability through its innovative design. AMVim strikes a balance between performance and stability, emerging as a powerful solution for vision tasks.

### 4.4 Semantic Segmentation

We evaluate AMVim on the downstream semantic segmentation task using the ADE20K dataset [42], with results summarized in Table 4. When integrated into the UperNet framework [43], AMVim consistently outperforms Vim across both full-precision and token-reduced settings. In the non-pruned setting, AMVim achieves improvements of +0.2% and +0.1% mIoU over Vim-Ti and Vim-S, respectively, demonstrating enhanced representational capacity due to its multi-scale scanning mechanism.

Under ToMe-based token reduction [18], the performance advantage of AMVim becomes even more pronounced, yielding mIoU gains of +5.9% and +6.7% over the corresponding pruned Vim models. This significant improvement under aggressive token merging highlights the robustness and generalization capability of AMVim in dense prediction tasks. These results confirm that AMVim not only improves accuracy in standard evaluation but also exhibits stronger generalization under token merging, making it a more effective and reliable backbone for efficient downstream vision tasks.

Table 4: Semantic segmentation performance on ADE20K [42] val set with UperNet. "ToMe" denotes token merging applied via ToMe [18] in a training-free manner. △ indicates the improvement of AMVim over the Vim baseline. AMVim consistently outperforms Vim, and when combined with ToMe, achieves significantly higher mIoU under heavy pruning.

| method | backbone | image size | #param (M) | val mIoU (%) | △ (%) |
|--------|----------|-----------|------------|-------------|-------|
| UperNet | Vim-Ti | $512^2$ | 13 | 40.0 | 0 |
| UperNet | Vim-S | $512^2$ | 46 | 43.3 | 0 |
| UperNet | AMVim-Ti | $512^2$ | 13 | **40.2** | 0.2↑ |
| UperNet | AMVim-S | $512^2$ | 46 | **43.4** | 0.1↑ |
| UperNet | ToMe-Vim-Ti | $512^2$ | 13 | 21.1 | 0 |
| UperNet | ToMe-Vim-S | $512^2$ | 46 | 22.0 | 0 |
| UperNet | ToMe-AMVim-Ti | $512^2$ | 13 | **27.0** | 5.9↑ |
| UperNet | ToMe-AMVim-S | $512^2$ | 46 | **28.7** | 6.7↑ |

### 4.5 Ablation Study

We conducted extensive ablation studies to validate the effectiveness of each component in AMVim. The scanning directions involved in these experiments are shown in Figure 5. This study specifically

Table 5: Comparison of Symmetric and Asymmetric Paths. △ denotes the performance gap between Vim (i.e., D1-D2) and other dual-path configurations. Asymmetric paths demonstrate superior pruning robustness than symmetric ones.

| path1 | path2 | reduction ratio | FLOPs (G) | top-1 acc. (%) | △ (%) |
|---|---|---|---|---|---|
| D1 | D2 | 0 | 1.45 | 76.1 | 0 |
| | | 0.17 | 1.27 | 38.7 | 0 |
| | | 0.27 | 1.13 | 41.6 | 0 |
| | | 0.36 | 1.00 | 37.6 | 0 |
| | | 0.46 | 0.86 | 34.2 | 0 |
| D1 | D4 | 0 | 1.45 | 76.0 | 0.1↓ |
| | | 0.17 | 1.27 | 55.9 | 17.2↑ |
| | | 0.27 | 1.13 | 54.5 | 12.9↑ |
| | | 0.36 | 1.00 | 48.6 | 11.0↑ |
| | | 0.46 | 0.86 | 44.9 | 10.7↑ |
| D1 | rand | 0 | 1.45 | 74.9 | 1.2↓ |
| | | 0.17 | 1.27 | 45.1 | 6.5↑ |
| | | 0.27 | 1.13 | 43.5 | 1.9↑ |
| | | 0.36 | 1.00 | 39.3 | 1.7↑ |
| | | 0.46 | 0.86 | 34.1 | 0.1↓ |

Table 6: Ablation study on the impact of the window-aware scanning mechanism on top-1 accuracy (%) during token reduction. Integrating the window-aware scanning mechanism significantly enhances resilience to pruning.

| path1 | path2 | reduction ratio | FLOPs (G) | top-1 acc. (%) | △ (%) |
|---|---|---|---|---|---|
| D1 | D2 | 0 | 1.45 | 76.1 | 0 |
| | | 0.17 | 1.27 | 38.7 | 0 |
| | | 0.27 | 1.13 | 41.6 | 0 |
| | | 0.36 | 1.00 | 37.6 | 0 |
| | | 0.46 | 0.86 | 34.2 | 0 |
| D1 | D2_D2 | 0 | 1.45 | 76.4 | 0.3↑ |
| | | 0.17 | 1.27 | 66.5 | 27.8↑ |
| | | 0.27 | 1.13 | 65.1 | 23.5↑ |
| | | 0.36 | 1.00 | 61.2 | 23.6↑ |
| | | 0.46 | 0.86 | 56.9 | 22.7↑ |
| D1 | D2_rand | 0 | 1.45 | 75.8 | 0.3↓ |
| | | 0.17 | 1.27 | 69.0 | 30.3↑ |
| | | 0.27 | 1.13 | 67.0 | 25.4↑ |
| | | 0.36 | 1.00 | 63.6 | 26.0↑ |
| | | 0.46 | 0.86 | 59.3 | 25.1↑ |
| D1 | D2_D4 | 0 | 1.45 | 76.3 | 0.2↑ |
| | | 0.17 | 1.27 | 72.8 | 34.1↑ |
| | | 0.27 | 1.13 | 70.3 | 28.7↑ |
| | | 0.36 | 1.00 | 66.7 | 29.1↑ |
| | | 0.46 | 0.86 | 61.3 | 27.1↑ |

focuses on dual-path configurations. For clarity, the hyphen symbol "-" represents the connection between the two paths, with the ends indicating the directions of the main paths. The underscore symbol "_" denotes the window-aware scanning mechanism, where the left end indicates the main path direction and the right end reflects the scanning direction within the window. Here, D1-D2 represents Vim [6], while D1-D2_D4 corresponds to AMVim. Additionally, the pruning-free performance discussed in this study is defined as the performance at a reduction ratio of 0.

**Effect of asymmetry.** Table 5 compares the performance of symmetric (D1-D2) and asymmetric paths (D1-D4 and D1-rand). While D1-D4 exhibits a marginal 0.1% decrease in pruning-free accuracy compared to D1-D2, it achieves consistent improvements of ≥10% across all reduction ratios, with a notable 17.2% gain at a reduction ratio of 0.17.

To further validate the importance of asymmetry, we introduce a random scanning direction in the second path (D1-rand). This configuration yields the lowest pruning-free performance, as the random path struggles to capture token relationships and impedes the training process. Despite its reduced pruning robustness relative to D1-D4, D1-rand still significantly outperforms the symmetric D1-D2. These results conclusively demonstrate that asymmetric path designs are essential for enhancing the pruning resilience of Mamba-based architectures.

**Effect of window-aware scanning.** Table 6 presents the ablation study results for the window-aware scanning mechanism, with global scanning directions aligned with Vim (D1-D2). For simplicity, this study focuses on applying the window-aware scanning mechanism to the second path, as similar trends are observed in the first path (see Table 10 in the Appendix).

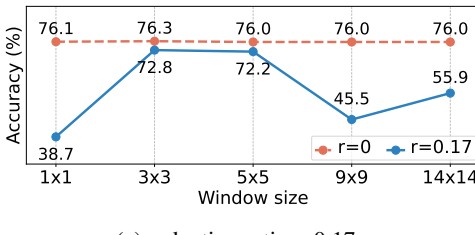

(a) reduction ratio = 0.17

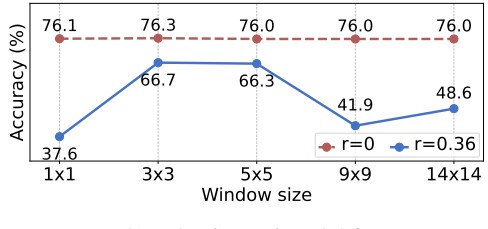

(b) reduction ratio = 0.36

Figure 6: Ablation study on the impact of window size on top-1 accuracy (%) during token reduction. The symbol 'r' represents the reduction ratio. Across different reduction ratios, moderate window sizes (i.e., 3×3 and 5×5) consistently demonstrate superior pruning resistance.

Table 7: Ablation study on the impact of multi-scale paths on top-1 accuracy (%) during token reduction. Red text in brackets indicates window sizes. The single-path window-aware design demonstrates superior robustness than the dual-path mechanism, despite both retaining asymmetric multi-scale properties.

| path1 | path2 | reduction ratio | FLOPs (G) | top-1 acc (%) | △ (%) |
|---|---|---|---|---|---|
| D1 | D2 | 0 | 1.45 | 76.1 | 0 |
| | | 0.17 | 1.27 | 38.7 | 0 |
| | | 0.27 | 1.13 | 41.6 | 0 |
| | | 0.36 | 1.00 | 37.6 | 0 |
| | | 0.46 | 0.86 | 34.2 | 0 |
| D1_D5(5) | D2_D4(3) | 0 | 1.45 | 75.2 | 0.8↓ |
| | | 0.17 | 1.27 | 69.9 | 31.3↑ |
| | | 0.27 | 1.13 | 66.6 | 25.0↑ |
| | | 0.36 | 1.00 | 62.8 | 25.3↑ |
| | | 0.46 | 0.86 | 58.1 | 23.9↑ |
| D1 | D2_D4 | 0 | 1.45 | 76.3 | 0.2↑ |
| | | 0.17 | 1.27 | 72.8 | 34.1↑ |
| | | 0.27 | 1.13 | 70.3 | 28.7↑ |
| | | 0.36 | 1.00 | 66.7 | 29.1↑ |
| | | 0.46 | 0.86 | 61.3 | 27.1↑ |

Table 8: Ablation study on the impact of window scanning direction on top-1 accuracy (%) during token reduction. The symbol △ quantifies performance differences relative to Vim (i.e., D1-D2). Distinct scanning directions between the window and global path ensure superior resistance to pruning.

| path1 | path2 | reduction ratio | FLOPs (G) | top-1 acc. (%) | △ (%) |
|---|---|---|---|---|---|
| D1 | D2 | 0 | 1.45 | 76.1 | 0 |
| | | 0.17 | 1.27 | 38.7 | 0 |
| | | 0.27 | 1.13 | 41.6 | 0 |
| | | 0.36 | 1.00 | 37.6 | 0 |
| | | 0.46 | 0.86 | 34.2 | 0 |
| D1 | D2_D2 | 0 | 1.45 | 76.4 | 0.3↑ |
| | | 0.17 | 1.27 | 66.5 | 27.8↑ |
| | | 0.27 | 1.13 | 65.1 | 23.5↑ |
| | | 0.36 | 1.00 | 61.2 | 23.6↑ |
| | | 0.46 | 0.86 | 56.9 | 22.7↑ |
| D1 | D2_D6 | 0 | 1.45 | 76.2 | 0.1↑ |
| | | 0.17 | 1.27 | 73.0 | 34.3↑ |
| | | 0.27 | 1.13 | 71.0 | 29.4↑ |
| | | 0.36 | 1.00 | 67.6 | 30.1↑ |
| | | 0.46 | 0.86 | 62.3 | 28.1↑ |
| D1 | D2_D4 | 0 | 1.45 | 76.3 | 0.2↑ |
| | | 0.17 | 1.27 | 72.8 | 34.1↑ |
| | | 0.27 | 1.13 | 70.3 | 28.7↑ |
| | | 0.36 | 1.00 | 66.7 | 29.1↑ |
| | | 0.46 | 0.86 | 61.3 | 27.1↑ |

The results reveal that introducing a same-direction window-aware scanning mechanism (D1-D2_D2) already significantly enhances pruning robustness, underscoring the importance of local dependency modeling in improving architectural robustness. When the main path and window scanning directions are further diversified (D1-D2_D4), performance improves by an additional 5% compared to D1-D2_D2, reinforcing the effectiveness of asymmetric design in boosting pruning resilience. Notably, the pruning-free performance increases by 0.3% for D1-D2_D2 and 0.2% for D1-D2_D4, highlighting the mechanism's ability to enhance model representation through local dependency capture.

When the window scanning direction is randomized (D1-D2_rand), its pruning robustness lies between D1-D2_D2 and D1-D2_D4, underscoring the irreplaceability of asymmetry and the importance of semantic relationship modeling. Additionally, the pruning-free performance of D1-D2_rand decreases by only 0.3%, demonstrating that the window mechanism effectively mitigates randomness by coordinating global and local information flows.

**Effect of window size.**

Window size is a critical hyperparameter introduced in this study. To assess its impact on performance, we conducted extensive ablation studies, as shown in Figure 6. In these experiments, the global path remains aligned with Vim (D1-D2), while the second path incorporates the window-aware scanning mechanism. Notably, a window size of $1\times1$ corresponds to the D1-D2 configuration, whereas a window size of $14\times14$ corresponds to the D1-D4 configuration.

The results reveal that the window size has a relatively minor effect on pruning-free performance, with optimal performance achieved at $3\times3$. However, token reduction performance exhibits significant fluctuations as the window size varies, with this trend remaining consistent across all reduction ratios (r = 0.17 in Figure 6a and r = 0.36 in Figure 6b). The optimal pruning robustness is observed at intermediate window sizes ($3\times3$ and $5\times5$), whereas excessively small or large windows all lead to a performance collapse.

This behavior is expected, as the window-aware mechanism is specifically designed to capture local dependencies. However, larger window sizes exacerbate Mamba's long-range forgetting issue [1, 35], reducing its effectiveness. Furthermore, the poor robustness of the $1\times1$ and $14\times14$ window sizes results from their deviation from the multi-scale path design. These findings demonstrate that careful window size selection is essential for harmonizing local dependencies with global context, a key factor in maximizing the efficacy of the multi-scale architecture.

**Effect of multi-scale path.** The proposed design integrates window-aware scanning into one path

while retaining sequential scanning in the other, forming asymmetric multi-scale paths. This raises a natural question: can a dual-path window-aware mechanism achieve comparable performance by varying window sizes?

As shown in Table 7, the dual-path window-aware configuration (D1_D5(5)-D2_D4(3)) underperforms the single-path window-aware design (D1-D2_D4), despite both maintaining asymmetric multi-scale properties. This performance gap arises from the homogeneous scanning mechanisms in dual window-aware paths. While varying window sizes introduce scale diversity, the lack of receptive field heterogeneity limits their ability to capture hierarchical spatial dependencies. In contrast, the single-path window-aware design achieves superior robustness through heterogeneous scanning mechanisms, balancing local granularity and global continuity.

**Effect of scan direction.** Table 8 summarizes ablation studies on the scanning direction within windows, with the global scanning direction fixed to align with Vim (D1-D2). The results show that when the window and main path share the same scanning direction (D1-D2_D2), accuracy drops by approximately 7% compared to heterogeneous scanning directions (D1-D2_D6 and D1-D2_D4). Although D1-D2_D2 maintains an asymmetric path design, its diversity in token connections remains significantly lower than that of the heterogeneous configurations, which primarily accounts for its inferior performance.

Additionally, we observe that as long as the scanning direction within the window differs from the main path, the performance remains largely comparable (i.e., D1-D2_D6 vs. D1-D2_D4). While D1-D2_D6 exhibits slightly better pruning robustness, D1-D2_D4 offers a marginal pruning-free performance improvement of 0.1%. In this study, we adopt D1-D2_D4 as our default architecture, but we will release weights for both configurations to allow flexibility in choice. These findings collectively demonstrate that pruning robustness is ensured by heterogeneous scanning directions between the window and main path, regardless of the specific direction chosen.

# 5 Conclusion

In this work, we propose AMVim, a vision Mamba architecture designed for pruning robustness. By incorporating a window-aware scanning mechanism into one of paths, AMVim enables asymmetric multi-scale scanning. Extensive experiments have verified the effectiveness and high robustness of AMVim, laying a foundation for future research on the stability of SSM-based models.
**Limitations.** Due to resource constraints, we have not yet fully explored the scalability of AMVim, such as validating its efficacy on multi-task scenarios or integrating it with cross-modal frameworks. Despite these limitations, our experiments conclusively demonstrate AMVim's superiority in pruning robustness and spatial modeling, and we plan to investigate its broader applicability in future work.

## Acknowledgments and Disclosure of Funding

This work was supported by National Major Scientific Instrument and Equipment Development Project of National Natural Science Foundation of China under Grant 62427820; in part by National Natural Science Foundation of China under Grant 62306198 and in part by Natural Science Foundation of Sichuan Province under Grant 2024NSFSC1468.

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

# A    Implement Details

AMVim is fine-tuned using the pretrained weights from Vim. The detailed training configurations are presented in Table 9. All other settings are aligned with those of Vim.

Table 9: Training settings for AMVim on ImageNet-1K.

| finetune config | AMVim-T | AMVim-S |
|---|---|---|
| optimizer | AdamW | AdamW |
| base learning rate | 4e-5 | 1e-5 |
| minimal learning rate | 1e-5 | 5e-6 |
| weight decay | 1e-8 | 0.05 |
| optimizer momentum | $\beta1,\beta2$=0.9,0.999 | $\beta1,\beta2$=0.9,0.999 |
| batch size | 1024 | 1024 |
| training epochs | 150 | 150 |
| learning rate schedule | cosine decay | cosine decay |
| warmup epochs | 5 | 5 |
| warmup learning rate | 1e-5 | 1e-5 |
| warmup schedule | linear | linear |
| drop path | 0 | 0.3 |
| mixup | 0.8 | 0.8 |
| cutmix | 1 | 1 |
| EMA | None | None |

Table 10: Comparison results with window-aware scanning mechanism applied on various paths. The window-aware mechanism on the second path exhibits stronger pruning resistance than on the first path.

| path1 | path2 | red. ratio | FLOPs(G) | top-1 acc(%) | $\triangle(\%)$ |
|---|---|---|---|---|---|
| | | 0 | 1.45 | 76.1 | 0 |
| | | 0.17 | 1.27 | 38.7 | 0 |
| | | 0.27 | 1.13 | 41.6 | 0 |
| | | 0.36 | 1.00 | 37.6 | 0 |
| D1 | D2 | 0.46 | 0.86 | 34.2 | 0 |
| | | 0 | 1.45 | 76.4 | 0.3↑ |
| | | 0.17 | 1.27 | 72.3 | 33.6↑ |
| | | 0.27 | 1.13 | 70.2 | 28.6↑ |
| | | 0.36 | 1.00 | 65.4 | 27.8↑ |
| D1_D3 | D2 | 0.46 | 0.86 | 56.8 | 22.6↑ |
| | | 0 | 1.45 | 76.3 | 0.2↑ |
| | | 0.17 | 1.27 | 72.8 | 34.1↑ |
| | | 0.27 | 1.13 | 70.3 | 28.7↑ |
| | | 0.36 | 1.00 | 66.7 | 29.1↑ |
| D1 | D2_D4 | 0.46 | 0.86 | 61.3 | 27.1↑ |

# B    More Experiments

**Impact of window-aware on various paths.** To further validate the superiority of our design, we applied the window-aware scanning mechanism to the first path (D1_D3-D2) for comparison. As shown in Table 10, both configurations exhibit comparable performance at reduction ratios of 0.17 and 0.27. However, as the reduction ratio increases, the D1-D2_D4 configuration significantly outperforms D1_D3-D2. This indicates that while the window-aware mechanism benefits either path, its integration into the second path yields optimal performance, particularly under aggressive pruning scenarios. **Impact of the global scanning direction.** In this work, we align the global scanning direction with Vim (i.e., D1-D2). To investigate the impact of global scanning direction, we conducted experiments by changing the global scanning direction of the second path from D2 to D4. As shown in Table 11, when the intra-window scanning direction is set to D2 (D1-D4_D2), pruning robustness improves but remains inferior to configurations with intra-window direction D6 (D1-D4_D6). This performance gap likely stems from the significant overlap in token connections and receptive fields between the D1 and D2 directions, limiting their complementary information flow.

Furthermore, while D1-D4_D6 achieves notable improvements in pruning robustness, it still underperforms our design with global direction D2 (D1-D2_D4) by approximately 2% in accuracy. These results demonstrate that the choice of global scanning direction is critical when combined with the window-aware mechanism.

**Impact of sequential scanning path.** To further investigate the role of the sequential scanning branch, we implemented the window-aware scanning mechanism on both paths. As shown in Table 12, regardless of the intra-window scanning direction, these configurations achieve comparable pruning robustness. While they significantly outperform Vim (which relies solely on sequential scanning) due to the stability introduced by the window-aware mechanism in capturing local dependencies, they exhibit performance degradation in pruning-free settings compared to Vim. This underscores the importance of sequential scanning for preserving base performance.

The proposed architecture combines sequential scanning with window-aware scanning, inheriting the strengths of both strategies. This combination achieves an optimal balance between pruning robustness and model performance.

Table 11: Ablation study on the impact of global scanning directions. △ denotes the performance gap relative to Vim (i.e., D1-D2). When integrated with the window-aware scanning mechanism, the global scanning direction D2 outperforms D4.

| path1 | path2 | red. ratio | FLOPs(G) | top-1 acc(%) | △(%) |
|---|---|---|---|---|---|
| D1 | D2 | 0 | 1.45 | 76.1 | 0 |
| | | 0.17 | 1.27 | 38.7 | 0 |
| | | 0.27 | 1.13 | 41.6 | 0 |
| | | 0.36 | 1.00 | 37.6 | 0 |
| | | 0.46 | 0.86 | 34.2 | 0 |
| D1 | D4_D2 | 0 | 1.45 | 76.2 | 0.1↑ |
| | | 0.17 | 1.27 | 59.7 | 21.1↑ |
| | | 0.27 | 1.13 | 59.2 | 17.6↑ |
| | | 0.36 | 1.00 | 54.6 | 17.0↑ |
| | | 0.46 | 0.86 | 51.1 | 17.0↑ |
| D1 | D4_D6 | 0 | 1.45 | 76.0 | 0.1↓ |
| | | 0.17 | 1.27 | 70.9 | 32.3↑ |
| | | 0.27 | 1.13 | 68.5 | 26.9↑ |
| | | 0.36 | 1.00 | 64.0 | 26.5↑ |
| | | 0.46 | 0.86 | 58.7 | 24.5↑ |
| D1 | D2_D4 | 0 | 1.45 | 76.3 | 0.2↑ |
| | | 0.17 | 1.27 | 72.8 | 34.1↑ |
| | | 0.27 | 1.13 | 70.3 | 28.7↑ |
| | | 0.36 | 1.00 | 66.7 | 29.1↑ |
| | | 0.46 | 0.86 | 61.3 | 27.1↑ |

Table 12: Ablation study on the impact of the sequential scanning path. △ represents the performance gap relative to Vim (i.e., D1-D2). While the dual-path window-aware design demonstrates superior pruning robustness compared to Vim, it suffers from performance degradation in pruning-free settings.

| path1 | path2 | red. ratio | FLOPs(G) | top-1 acc(%) | △(%) |
|---|---|---|---|---|---|
| D1 | D2 | 0 | 1.45 | 76.1 | 0 |
| | | 0.17 | 1.27 | 38.7 | 0 |
| | | 0.27 | 1.13 | 41.6 | 0 |
| | | 0.36 | 1.00 | 37.6 | 0 |
| | | 0.46 | 0.86 | 34.2 | 0 |
| D1_D1 | D2_D2 | 0 | 1.45 | 75.8 | 0.3↓ |
| | | 0.17 | 1.27 | 69.8 | 31.2↑ |
| | | 0.27 | 1.13 | 67.1 | 25.5↑ |
| | | 0.36 | 1.00 | 63.2 | 25.6↑ |
| | | 0.46 | 0.86 | 58.3 | 24.2↑ |
| D1_D3 | D2_D4 | 0 | 1.45 | 75.7 | 0.4↓ |
| | | 0.17 | 1.27 | 70.3 | 31.6↑ |
| | | 0.27 | 1.13 | 66.6 | 25.0↑ |
| | | 0.36 | 1.00 | 61.8 | 24.3↑ |
| | | 0.46 | 0.86 | 56.3 | 22.1↑ |
| D1_D3 | D2_D6 | 0 | 1.45 | 75.4 | 0.6↓ |
| | | 0.17 | 1.27 | 70.6 | 31.9↑ |
| | | 0.27 | 1.13 | 67.5 | 25.9↑ |
| | | 0.36 | 1.00 | 63.5 | 25.9↑ |
| | | 0.46 | 0.86 | 58.3 | 24.2↑ |

