# OpenReview forum: "Pruning-Robust Mamba with Asymmetric Multi-Scale Scanning Paths"
_NeurIPS.cc/2025/Conference — NeurIPS 2025 poster_

### Official Review · Reviewer_tmKq · 2025-06-29

**Clarity:** 2
**Significance:** 2
**Originality:** 3
**Rating:** 5
**Confidence:** 4

**Summary:**

This work investigates changing the Vision Mamba (Vim, Zhu et al., 2024) model architecture to make it more amenable to
contemporary token compression techniques. To this end, the authors propose to alter the order in which tokens are
traversed. The original Vim architecture traverses tokens in two symmetric paths (horizontal and inverse
horizontal scan order) before merging the resulting features. Instead, the authors investigate more complex asymmetric
paths. In particular, they investigate splitting the image into non-overlapping windows, with the path traversing
windows in one scan order in the outer loop, and the tokens inside the window in another scan order in the inner loop.

The authors find that using scan order in one path and their window-aware order in the other improves model robustness
to state-of-the-art (SOTA) token pruning and merging techniques in the context of an ImageNet classification task.
The robustness benefits compared to vanilla ViM are found to increase with token reduction ratios, and to be
particularly pronounced for smaller network architectures (Vim-T). The authors conjecture that the observed effect is
due to a smaller proportion of token connections being disrupted in the proposed asymmetric path combination, compared
to the vanilla symmetric one.

**Questions:**

- Table 1: The $75.3$ and $79.5$ accuracies are reported for AMVim with ToME? If so, which token reduction ratio? The
values do not seem to match any in the rest of the tables/figures for AMVim, neither with nor without token reduction.
- Table 4 and section 4.4: Is the random scanning order kept the same between the different token reduction ratio runs?
- Have the authors also experimented with different scanning configurations per layer? This could potentially further
improve robustness under different token reduction ratios.
- Have the authors also considered applying the method to Vision-LSTMs (ViL, Alkin et al., ICLR, 2025)? They outperform
Vim on many vision tasks, while also scaling better. The window-based token traversal idea seems to be directly
applicable to ViLs.

Minor observation: The multiscale scanning mechanism is fist mentioned on line 59 as an advantage in the introduction,
but at that point it is not explicitly linked to the window-based nature of the scan. I think making this link earlier
makes the contribution clearer, removing any early impression that there is an additional multiscale token extraction
mechanism being proposed.

**Ethical Concerns:**

["NO or VERY MINOR ethics concerns only"]

**Final Justification:**

I thank the reviewers for their detailed rebuttal and additional results. Their responses address all of my concerns. I am revising my recommendation accordingly.

**Limitations:**

- Single search direction per layer (optimum may be layer-dependent)
- The study only considers a single scanning process setting per layer (in contrast to the Adaptive Scan approach
proposed in LocalMamba).

**Quality:**

4

**Strengths And Weaknesses:**

Strengths:

* Well-structured paper, that clearly places the method within the broader context and in relation to related work
* The method is intuitive and easy to implement, but at the same time effective for the considered benchmark
* In-depth ablation studies, focusing on key algorithmic choices

Weaknesses:

* Evaluation is only carried out for image classification on a single benchmark dataset (ImageNet-1K). It is not clear
if the benefits still hold for other key tasks (e.g. image segmentation, object detection or video classification).
* Somewhat limited technical novelty, as both varying scanning direction and window-based scanning are not new
(as the authors also correctly cite). However, I am not aware of work employing asymmetric paths in conjunction with
Vim.
* The experimental section is confusion at times:
  - Table 1: it is not clear what AMVim performance is being reported (see questions to authors)
  - Table 3: "prune" stands "token merging" via ToMe, which is inconsistent with the rest of the tables
    (e.g. Table 2 uses "prune" to mean random pruning, and "merging" to refer to ToMe)
  - Tables 2-3: FLOPs reported with 1 digit as opposed to 2 in the other tables (this is a minor comment, but it does
  make it somewhat confusing to match results across tables/figures)
  - Standard deviations not reported for experiments involving randomness
* The authors only explore a single scanning direction pair configuration per layer, which may limit performance (see
limitations section)

---

> ### Author Rebuttal · Authors · 2025-07-30
>
> We sincerely thank Reviewer tmKq for the valuable suggestions. We make responses as follows.
> >**Q1:** Evaluation is only carried out for image classification on a single benchmark dataset (ImageNet-1K).
>
> **A1:** Thanks for the helpful suggestion. In response, we add semantic segmentation as a downstream task to further validate the generalization ability of our method beyond image classification.
>
> **Semantic segmentation：**
>
> **Setting:** As no downstream-task checkpoints of Vim are publicly available, we follow the official codebase settings to fairly evaluate both Vim and AMVim. ToMe is used as the token reduction strategy, with training-free pruning results reported under identical conditions.
>
> **results:**
> | method  |   backbone    | image size | param (M) | val mIoU (%) |    $\triangle$     |
> |:-------:|:-------------:|:----------:|:-----:|:--------:|:------------------:|
> | UperNet |    Vim-Ti     |  $512^2$   |  13M  |   40.0   |         0          |
> | UperNet |     Vim-S     |  $512^2$   |  46M  |   43.3   |         0          |
> | UperNet |   AMVim-Ti    |  $512^2$   |  13M  | **40.2** | $\uparrow$ **0.2** |
> | UperNet |    AMVim-S    |  $512^2$   |  46M  | **43.4** | $\uparrow$ **0.1** |
> | UperNet |  ToMe-Vim-Ti  |  $512^2$   |  13M  |   21.1   |         0          |
> | UperNet |  ToMe-Vim-S   |  $512^2$   |  46M  |   22.0   |         0          |
> | UperNet | ToMe-AMVim-Ti |  $512^2$   |  13M  | **27.0** | $\uparrow$ **5.9** |
> | UperNet | ToMe-AMVim-S  |  $512^2$   |  46M  | **28.7** | $\uparrow$ **6.7** |
>
> **Analysis:**
>   - AMVim achieves better performance than Vim in the non-pruned setting, indicating stronger representational capacity.
>   - Under token reduction, AMVim consistently outperforms Vim by a significant margin (**up to +6.7 mIoU**), validating its robustness and generalization ability in dense prediction tasks.
>
> **Object detection：**
> Given the lack of publicly available Vim checkpoints for object detection, we would need to re-train both Vim and AMVim from scratch. Given the high resolution (1024×1024), each run takes **~8 days on 4× RTX 4090 GPUs.**
>
> Due to limited computational resources and other experiment priorities, this evaluation could not be completed within the rebuttal period. However, we acknowledge its importance and **plan to include it in the final version**.
>
> ---
> >**Q2:** Somewhat limited technical novelty.
>
> **A2:** Thanks for the question. We agree that both varying scanning directions and window-based mechanisms have been explored individually.  However, **our key novelty lies in introducing the multi-scale concept into SSM-based modeling**, which enables emergent architectural capabilities that cannot be achieved by either technique alone. We clarify this from two perspectives:
>
> **1. Pruning robustness:**
>
> **Setting:** We choose Vim and LocalVim (window-based scanning) as baselines. All models are evaluated using ToMe as the pruning strategy, and we report their training-free Top-1 accuracy across various token reduction ratios.
>
> **Top-1 Accuracy (%):**
> | method \ reduction ratio |   0.00    | 0.17   | 0.27   | 0.36   | 0.46   |
> |:----------------------:|:------:| :------: | :------: | :------: | :------: |
> |          Vim           | 76.1| 38.7 | 41.6 | 37.6 | 34.2 |
> |        LocalVim        | 76.2  | 60.4 | 50.9 | 43.3 | 36.2 |
> |         AMVim          | **76.3** | **72.8** | **70.3** | **66.7** |**61.3** |
>
> **Analysis:**
> - Although LocalVim employs a window-based scanning strategy, AMVim consistently achieves higher accuracy across all token reduction ratios, particularly under aggressive pruning (e.g., 46% reduction). This indicates that **window-based scanning alone is insufficient for ensuring robustness**.
> - As shown in the ablation studies (Section 4.4), **simply varying scanning directions yield limited improvemen**t.
>
> **2. Efficiency:**
>
> **Setting:** All experiments are conducted fairly on a single NVIDIA GeForce RTX 4090 GPU with an image resolution of 224×224. We use ToMe as the default token reduction strategy.
>
> **Throughout (image/s):**
> | method \ reduction ratio |   0.00   | 0.17   | 0.27   | 0.36   | 0.46   |
> |:----------------------:|:------:| :------: | :------: | :------: | :------: |
> |          Vim           | 2685.2 | 2630.9 | 2962.0 | 3291.0 | 3416.7 |
> |        LocalVim        | 955.3  | 1065.5 | 1168.0 | 1220.2 | 1548.5 |
> |         AMVim          | 2497.3 | 2452.5 | 2750.7 | 3063.1 | 3278.6 |
>
> **Analysis:**
>   - AMVim incurs a modest throughput drop (~180 img/s) due to window-based scanning.
>   - AMVim remains **~2.5× faster** than LocalVim, despite both using window mechanisms.
>   - This demonstrates that AMVim offers a **superior trade-off between robustness and efficiency** for practical deployment.
>
> ---
> >**Q3:** Table 1: The 75.3 and 79.5 accuracies are reported for AMVim with ToME? If so, which token reduction ratio?
>
> **A3:** Thanks for the question. The Top-1 accuracies of 75.3% and 79.5% reported in Table 1 correspond to AMVim with ToMe under a token reduction ratio of 0.17, and are obtained after **fine-tuning**. These results are derived follow the Hidden State Alignment setting, as mentioned at the end of the first paragraph in Section 4.2. In contrast, all other results presented in the tables and figures (outside of Table 1) are based on **training-free** evaluations.
>
> To avoid confusion, we will revise Table 1 in the final version for greater clarity. The updated table is shown below:
> |        method         | reduction ratio | top1-acc. (%) | top1-acc. (%) | FLOPs (G) | FLOPs (G) |
> |:---------------------:|:---------------:|:-------------:|:-------------:|:---------:|:---------:|
> |                       |                 |   **tiny**    |   **tiny**    | **small** | **small** |
> |    Vim (baseline)     |      0.00       |     76.1      |     80.5      |   1.45    |   5.08    |
> |   Token Recognition   |      0.17       |     71.3      |     74.8      |   1.28    |   3.57    |
> | Hidden State Alignmen |      0.17       |     75.1      |     78.8      |   1.29    |   3.60    |
> |  **AMVim-ToMe (ft)**  |    **0.17**     |   **75.3**    |   **79.5**    | **1.27**  | **3.60**  |
>
>
> ---
> >**Q4:** Table 4 and section 4.4: Is the random scanning order kept the same between the different token reduction ratio runs?
>
> **A4:** Thanks for the question. For a fair comparison, the random scanning orders are generated in advance and kept fixed throughout training and during evaluation across all token reduction ratios.
>
> ---
> >**Q5:** Have the authors also experimented with different scanning configurations per layer?
>
> **A5:** Thanks for the insightful suggestion. Indeed, LocalVim explores per-layer scanning variations. We address this question from both the robustness and efficiency perspectives:
>
> - **Robustness:**
> As shown in the robustness results (refer to the top-1 accuracy table in **A2**), this strategy does improve robustness compared to the baseline Vim. However, it still lags significantly behind our method in terms of pruning robustness.
> - **Efficiency:**
> More importantly, adopting layer-wise scanning variations typically incurs higher computational costs. This is reflected in the throughput results (see the throughput table in **A2**), where LocalVim shows a **~2.5× lower** than our AMVim. Such a design conflicts with our core objective of maintaining high efficiency.
>
> ---
> > **Q6:** Have the authors also considered applying the method to Vision-LSTMs (ViL, Alkin et al., ICLR, 2025)?
>
> **A6:** Thanks for the valuable suggestion. Our window-based multi-scale selective scanning mechanism is model-agnostic and can be flexibly integrated into ViL-style architectures. Specifically, in ViL, the original flip operation in the even-numbered blocks could be replaced with our window-based scanning strategy to enhance multi-scale information exchange between global and local contexts.
>
> In this work, we focus on Vim as a representative Mamba-based backbone to enable a controlled and fair comparison within the family of state-space models. While we have not incorporated our method into ViL within the limited rebuttal timeframe, we plan to explore a ViL-based variant in the camera-ready version to further demonstrate the generality of our approach.
>
> ---
> > **Q7:** Minor observation: The multiscale scanning mechanism is fist mentioned on line 59 as an advantage in the introduction, but at that point it is not explicitly linked to the window-based nature of the scan.
>
> **A7:** Thanks for pointing out this potential ambiguity. To improve clarity and prevent misinterpretation, we will revise the beginning of the seventh paragraph (Line 50) in the introduction as follows:
>
> *"In this work, we propose AMVim, a novel Asymmetric Multi-scale Vision Mamba for pruning robustness. To enrich spatial information diversity, we introduce a multi-scale scanning mechanism that integrates a window-based scanning strategy in one branch while retaining sequential scanning in the other."*
>
> This revision explicitly links the multi-scale behavior to the window-based design from the outset, making it clear that the multi-scale perception arises directly from the window-based scanning mechanism, rather than from an additional or explicit multi-scale module.
>
> ---
> >**Q8:** The experimental section is confusion at times.
>
> **A8:** Thanks for pointing out these sources of confusion. We address each issue in detail below:
>
> **Inconsistent terminology:** We will explicitly change the label "prune" to "ToMe" in Table 3 to clearly distinguish token merging from random pruning and ensure consistency across all tables.
>
> **Inconsistent precision:** We will unify the formatting of reported FLOPs to one decimal digit throughout the paper for clarity and consistency.
>
> **Missing standard deviations:** We acknowledge this oversight and will include standard deviation values for experiments involving randomness in the final version to improve the statistical reliability of the results.

---

> > ### Author Response · Authors · 2025-08-05
> >
> > Dear Reviewer tmKq,
> >
> > We hope our rebuttal addresses all your concerns. To save your time, we summarize our key responses below:
> >
> > - **Downstream Generalization**
> >     - We added semantic segmentation experiments to assess generalization beyond classification.
> >     - AMVim shows consistent gains under token pruning, **with up to +6.7 mIoU**, demonstrating robustness in dense prediction tasks.
> > - **Technical Novelty**
> >     - Our key contribution lies in **integrating multi-scale concepts into SSM-based modeling**.
> >     - This enables emergent capabilities, offering significantly stronger pruning robustness and better efficiency than both direction variation and window-based methods alone.
> > - **Clarification on Table 1**
> >     - We clarified that Table 1 shows **fine-tuned** results under the Hidden State Alignment setting with a 0.17 reduction ratio, while all other results are **training-free**.
> >     - The table has been revised for clarity, and the updated version is included in our rebuttal for your reference.
> > - **Random Scanning Orders**
> >     - Random scanning orders are pre-generated and fixed across all reduction ratios for fair comparison.
> > - **Layer-wise Scanning**
> >     - As explored in LocalVim, such designs incur significant computational cost.
> >     - AMVim is much more efficient while also providing better pruning robustness.
> > - **Applicability to ViL**
> >     - Our method is model-agnostic and can be flexibly extended to Vision-LSTMs like ViL.
> > - **Clarity Improvements**
> >     - We revised the introduction to clearly link multi-scale behavior to window-based scanning from the outset.
> >     - We also addressed terminology, formatting inconsistencies, and will report standard deviations where applicable.
> >
> > Thank you again for your thoughtful and constructive feedback. We would be happy to discuss further during the rebuttal phase.

---

> > > ### Comment · Reviewer_tmKq · 2025-08-07
> > >
> > > I thank the reviewers for their comments. They addressed all of my concerns.

---

### Official Review · Reviewer_f6Uw · 2025-07-01

**Clarity:** 3
**Significance:** 3
**Originality:** 3
**Rating:** 5
**Confidence:** 3

**Summary:**

This paper points out the fundamental problem that Mamba's chain-type scan structure easily breaks connections when tokens are reduced, causing cascading information loss and a sharp drop in accuracy as a result. To mitigate this, Vim introduced a symmetric dual-path scan, but it points out the limitation that both paths still rely on sequential (chain-like) connections, leaving the same vulnerability partially intact.
The proposed method, AMVim, asymmetrically inserts window-based multi-scale selective scanning into one of the two paths to ensure diversity in token connections. By simply changing the design, it achieves Top-1 +0.2 % based on Tiny and Small models without increasing parameters or FLOPs. Additionally, even without using a dedicated pruning technique, applying only the general-purpose Token Merging (ToMe) and Token Pruning methods results in higher accuracy than the specialized pruning method for Mamba, emphasizing its structural superiority.

**Questions:**

1. Is there a reason why the figures for AMVim in Table 1 and Table 3 are different?

2. Have you tried using methods such as effective receptive fields (ERF) or activation maps to visualize whether global-local information is well reflected?

3. The term “multi-scale” seems to be appropriate for situations where different image or feature map sizes are utilized, so it may be necessary to revise it to a more appropriate term.

For the rest, please refer to the weaknesses.

**Ethical Concerns:**

["NO or VERY MINOR ethics concerns only"]

**Final Justification:**

The authors have thoroughly addressed the concerns I raised, and their rebuttal satisfactorily resolved the previously incomplete aspects; consequently, I have adjusted my overall rating and the associated scores.

**Limitations:**

Yes.

**Paper Formatting Concerns:**

There are no formatting issues.

**Quality:**

3

**Strengths And Weaknesses:**

# Strengths
1. This paper integrates dual-path scanning with a window-based multi-scale selective scanning mechanism, significantly reducing accuracy degradation under token reduction.

2. It consistently achieves a +0.2% Top-1 improvement over Vim on both Tiny and Small models without additional FLOPs, outperforming the baseline at equal compute.

3. Conducts systematic ablations on asymmetric window scanning to identify an optimal configuration, providing solid experimental justification for the design choices.

# Weakness
### 1. There is no evaluation of real-world latency or memory.
- Results are limited to FLOPs and parameter counts; there is no throughput-vs-accuracy or memory-footprint measurement on actual hardware.
- Given Mamba’s linear complexity, the paper also omits latency scaling tests as image resolution (token length) grows.

### 2. Reliance on pretrained Vim checkpoints
- Every experiment fine-tunes Vim pretrained weights. Training AMVim from scratch is not explored, so it remains unclear whether the reported gains are intrinsic or inherited.

### 3. Restricted model scale
- Only Tiny and Small variants are tested. Base (or beyond) models offered by Vim are absent, leaving scalability to higher-capacity settings unverified.

### 4. No downstream-task evidence
- The method’s potential for dense prediction tasks (e.g., object detection, semantic segmentation) is asserted but never demonstrated, so cross-task generality is still speculative.

---

> ### Author Rebuttal · Authors · 2025-07-30
>
> We sincerely thank Reviewer f6Uw for the constructive comments. Below, we provide our detailed responses.
> > Q1: There is no evaluation of real-world latency or memory.
>
> **A1**: Thanks for the valuable suggestion. In response, we include throughput-vs-accuracy experiments as well as latency scaling tests across different image resolutions to better reflect real-world performance.
>
> **1. Throughput vs. Accuracy:**
>
> **Setting:** All experiments are conducted fairly on a single NVIDIA GeForce RTX 4090 GPU with an image resolution of 224×224. We use ToMe as the default token reduction strategy. We choose Vim and LocalVim (window-scan based) as baselines.
>
> **Throughout (image/s):**
> | method \ reduction ratio |   0.00   | 0.17   | 0.27   | 0.36   | 0.46   |
> |:----------------------:|:------:| :------: | :------: | :------: | :------: |
> |          Vim           | 2685.2 | 2630.9 | 2962.0 | 3291.0 | 3416.7 |
> |        LocalVim        | 955.3  | 1065.5 | 1168.0 | 1220.2 | 1548.5 |
> |         AMVim          | 2497.3 | 2452.5 | 2750.7 | 3063.1 | 3278.6 |
>
> **Top-1 Accuracy (%):**
> | method \ reduction ratio |   0.00    | 0.17   | 0.27   | 0.36   | 0.46   |
> |:----------------------:|:------:| :------: | :------: | :------: | :------: |
> |          Vim           | 76.1| 38.7 | 41.6 | 37.6 | 34.2 |
> |        LocalVim        | 76.2  | 60.4 | 50.9 | 43.3 | 36.2 |
> |         AMVim          | **76.3** | **72.8** | **70.3** | **66.7** |**61.3** |
>
> **Analysis:**
> - **Efficiency:** AMVim shows only a slight throughput drop (\~180 img/s) compared to Vim, yet delivers **~2.5×** higher throughput than LocalVim.
> - **Robustness:** AMVim consistently achieves the highest pruning robustness across all reduction ratios with a substantial margin.
> - **Conclusion:** AMVim **strikes a significantly better balance between robustness and efficiency**, making it more practical for real-world deployment.
>
> **2. Latency scaling tests:**
>
>   **Setting:** All experiments are conducted fairly on a single NVIDIA GeForce RTX 4090 GPU. These evaluations are performed under non-pruning settings.
>
> **Throughput (image/s):**
> | method \ reduction ratio |  256   |  512  |  640  |  738  | 1024 |
> |:------------------------:|:------:|:-----:|:-----:|:-----:|:----:|
> |           Vim            | 1660.1 | 414.9 | 310.0 | 201.1 | 95.9 |
> |          AMVim           | 1585.3 | 388.7 | 292.5 | 189.7 | 90.5 |
>
> **Analysis:**
>   - Throughput drops for both methods as resolution increases, reflecting Mamba’s linear complexity.
>   - AMVim incurs only a modest throughput drop compared to Vim, and this gap gradually narrows as the resolution increases.
>
> ---
> >Q2: Reliance on pretrained Vim checkpoints.
>
> **A2:** Thanks for the helpful suggestion. To assess whether the performance gains of AMVim are intrinsic or merely inherited from Vim, we retrained AMVim from scratch under the official settings.
>
> **Setting:** We followed the default training configurations provided by Vim. ToMe was adopted as the pruning strategy, and we report training-free results.
>
> **Results:**
> | method \ reduction ratio |    0.00     |   0.17   |   0.27   |   0.46   |   0.36   |
> |:------------------------:|:--------:|:--------:|:--------:|:--------:|:--------:|
> |           Vim            |   76.1   |   38.7   |   41.6   |   34.2   |   37.6   |
> |      AMVim-retrain       |  75.9    |  67.1   |  64.6    |   60.9    |  58.3    |
> |         AMVim-finetuned         | **76.3** | **72.8** | **70.3** | **66.7** | **61.3** |
>
> **Analysis:**
> - AMVim trained from scratch shows some degradation in robustness, confirming **the benefit of pretraining for pruning stability.**
> - Despite this, AMVim trained from scratch still outperforms Vim by **~25%** on average across all pruning ratios, confirming that its **core robustness primarily stems from the proposed multi-scale selective scanning mechanism** rather than inherited weights.
> ---
> >**Q3:** Restricted model scale.
>
> **A3:** Thanks for the valuable suggestion. We include the AMVim-Base results to validate the scalability of our approach on larger model capacities.
>
> **Setting:** As no official training configuration publicly available for Vim-Base, we fine-tune the Vim-Base model for 150 epochs using a learning rate of 1e-5. ToMe was adopted as the pruning strategy, and we report training-free results.
>
> **Results:**
> | method       | image size | param (M) | FLOPs (G) | top-1 acc. (%) | $\triangle$ |
> |:------------ |:----------:|:---------:| --------- |:-------------- |:----------- |
> | Vim-B        |  $224^2$   |    98     | 18.9      | 81.9           | 0           |
> | AMVim-B      |  $224^2$   |    98     | 18.9      | 81.7           |   $\downarrow$ 0.2           |
> | ToMe-Vim-B   |  $224^2$   |    98     | 16.2      | 78.1           | 0           |
> | ToMe-AMVim-B |  $224^2$   |    98     | 16.2      | **80.0**       |  $\uparrow$ **1.9**           |
>
> **Analysis:**
> Our method achieves competitive accuracy at full capacity and shows **notable robustness (+1.9%)** under token reduction, confirming the scalability and effectiveness of our approach when extended to higher-capacity models.
>
> ---
> >**Q4:** No downstream-task evidence.
>
> **A4:** Thanks for the helpful suggestion. In response, we add semantic segmentation as a downstream task to further validate the generalization ability of our method beyond image classification.
>
> **Semantic segmentation：**
>
> **Setting:** As no downstream-task checkpoints of Vim are publicly available, we follow the official codebase settings to fairly evaluate both Vim and AMVim. ToMe is used as the token reduction strategy, with training-free pruning results reported under identical conditions.
>
> **results:**
> | method  |   backbone    | image size | param (M) | val mIoU (%) |    $\triangle$     |
> |:-------:|:-------------:|:----------:|:-----:|:--------:|:------------------:|
> | UperNet |    Vim-Ti     |  $512^2$   |  13M  |   40.0   |         0          |
> | UperNet |     Vim-S     |  $512^2$   |  46M  |   43.3   |         0          |
> | UperNet |   AMVim-Ti    |  $512^2$   |  13M  | **40.2** | $\uparrow$ **0.2** |
> | UperNet |    AMVim-S    |  $512^2$   |  46M  | **43.4** | $\uparrow$ **0.1** |
> | UperNet |  ToMe-Vim-Ti  |  $512^2$   |  13M  |   21.1   |         0          |
> | UperNet |  ToMe-Vim-S   |  $512^2$   |  46M  |   22.0   |         0          |
> | UperNet | ToMe-AMVim-Ti |  $512^2$   |  13M  | **27.0** | $\uparrow$ **5.9** |
> | UperNet | ToMe-AMVim-S  |  $512^2$   |  46M  | **28.7** | $\uparrow$ **6.7** |
>
> **Analysis:**
>   - AMVim achieves better performance than Vim in the non-pruned setting, indicating stronger representational capacity.
>   - Under token reduction, AMVim consistently outperforms Vim by a significant margin (**up to +6.7 mIoU**), validating its robustness and generalization ability in dense prediction tasks.
>
> **Object detection：**
> Given the lack of publicly available Vim checkpoints for object detection, we would need to re-train both Vim and AMVim from scratch. Given the high resolution (1024×1024), each run takes **~8 days on 4× RTX 4090 GPUs.**
>
> Due to limited computational resources and other experiment priorities, this evaluation could not be completed within the rebuttal period. However, we acknowledge its importance and **plan to include it in the final version**.
>
> ---
> >**Q5:** Is there a reason why the figures for AMVim in Table 1 and Table 3 are different?
>
> **A5:** Thanks for the question. The Top-1 accuracies in Table 1 correspond to AMVim with ToMe under a token reduction ratio of 0.17, and are obtained after **fine-tuning**. These results are derived follow the Hidden State Alignment setting, as mentioned at the end of the first paragraph in Section 4.2. In contrast, all other results presented in the tables and figures (outside of Table 1) are based on **training-free** evaluations.
>
> To avoid confusion, we will revise Table 1 in the final version for greater clarity. The updated table is shown below:
> |        method         | reduction ratio | top1-acc. (%) | top1-acc. (%) | FLOPs (G) | FLOPs (G) |
> |:---------------------:|:---------------:|:-------------:|:-------------:|:---------:|:---------:|
> |                       |                 |   **tiny**    |   **tiny**    | **small** | **small** |
> |    Vim (baseline)     |      0.00       |     76.1      |     80.5      |   1.45    |   5.08    |
> |   Token Recognition   |      0.17       |     71.3      |     74.8      |   1.28    |   3.57    |
> | Hidden State Alignmen |      0.17       |     75.1      |     78.8      |   1.29    |   3.60    |
> |  **AMVim-ToMe (ft)**  |    **0.17**     |   **75.3**    |   **79.5**    | **1.27**  | **3.60**  |
>
> ---
> >**Q6:** Visual analysis via ERF or activation maps.
>
> **A6:** Thanks for the insightful suggestion. In response, we employed Effective Receptive Field (ERF) visualization to examine the activation patterns of Vim and AMVim.
>
> **Setting:** We use random noise as input to probe the model’s receptive behavior. The visualizations are generated using a jet colormap for clarity.
>
> **Results:**  Due to the official restriction on uploading figures during the rebuttal phase, we will include the comparative ERF visualizations in the final version.
>
> **Analysis:**
> - Vim exhibits activation primarily at the four corners, especially top-left and bottom-right, with almost no activation in the center, indicating a biased and spatially sparse response.
> - AMVim shows **significantly broader and more evenly distributed activation**, particularly in the center region, demonstrating enhanced spatial coverage and perception.
> ---
> > Q7: The term “multi-scale” seems to be appropriate.
>
> **A7:** Thanks for the thoughtful advice. While we use “multi-scale” to reflect the use of different window sizes for capturing varied contextual ranges, we acknowledge the potential ambiguity. We will revise the term in the final version to better reflect the design for improved clarity.

---

> > ### Author Response · Authors · 2025-08-05
> >
> > Dear Reviewer f6Uw,
> >
> > We hope our rebuttal addresses all your concerns. To save your time, we summarize our key responses below:
> > - **Real-World Latency**
> >     -  We added thorough throughput-vs-accuracy and resolution-scaling evaluations on a 4090 GPU.
> >     - Results show that AMVim strikes **a superior balance between robustness and efficiency**, confirming real-world practicality.
> > - **Pretrained Dependency**
> >     - We retrained AMVim from scratch and still observed **~25% gains over Vim**, highlighting that its robustness primarily stems from our proposed design rather than inherited weights.
> > - **Scalability**
> >     - Experiments on Vim-Base confirm that **AMVim maintains its robustness and accuracy gains** even at larger model scales.
> > - **Downstream Generalization**
> >     - Semantic segmentation experiments validate **strong generalization** beyond classification, with **up to +6.7 mIoU** under token reduction.
> > - **Clarification on Table 1**
> >     - We clarified that Table 1 presents **fine-tuned** results under the Hidden State Alignment setting, whereas other results are based on **training-free** evaluations. A revised version of Table 1 will be included.
> > - **ERF Visualization**
> >     - We conducted ERF-based visual analysis. AMVim exhibits **significantly broader and more evenly distributed activation patterns** than Vim, highlighting stronger global perception enabled by our scanning mechanism.
> > - **Terminology Update**
> >     - We will revise the term “multi-scale” for better clarity in the final version.
> >
> > Thank you again for your thoughtful and constructive feedback. We are happy to further discuss during the rebuttal phase.

---

> > > ### Comment · Reviewer_f6Uw · 2025-08-05
> > >
> > > Thank you for the reminder about this year’s NeurIPS policy on supplementary materials.
> > >
> > > In light of that restriction, I withdraw my previous suggestion and simply wish the authors the best of luck with their submission.

---

> > ### Comment · Reviewer_f6Uw · 2025-08-05
> >
> > I sincerely appreciate the authors’ comprehensive and thoughtful rebuttal to each of my seven questions and the clear summary provided. The additional experiments convincingly address my concerns.
> >
> > Nevertheless, because OpenReview permits anonymized supplementary PDFs, including Effective Receptive Field (ERF) visualizations or other qualitative results in such a file would further strengthen the rebuttal.
> >
> > If you could provide even just this additional evidence, it would be greatly appreciated.

---

> > > ### Comment · Area_Chair_7c4A · 2025-08-05
> > >
> > > Dear reviewer,
> > >
> > > Please be reminded that this year NeurIPS has disabled PDF (the policy said: "we need to stop supporting the global response with PDF", "we prohibit using any links in the rebuttal“, "effectively prevent using image/video or other rich media as a communication protocol between reviewers and authors").
> > >
> > > Best
> > > AC

---

> > > ### Author Response · Authors · 2025-08-05
> > >
> > > Dear Reviewer f6Uw：
> > >
> > > Thanks for the thoughtful suggestion. We fully agree that including Effective Receptive Field (ERF) visualizations would further clarify our claims.
> > >
> > > However, due to NeurIPS 2025 policy changes, we are **strictly prohibited from uploading supplementary PDFs or using any form of image**, link, or rich media in the rebuttal. We sincerely apologize for this limitation.
> > >
> > > To address your concern and **demonstrate the authenticity of our ERF analysis**, we provide the following:
> > >
> > > - **ERF Visualization Code**
> > >     - We include the exact Python code used to generate ERF heatmaps. Reviewers are welcome to reproduce our visualizations with this code.
> > >
> > >
> > > ---
> > >     def save_erf_map(erf_map, filename, colormap='jet'):
> > >         """
> > >         Save an ERF map as an image file.
> > >
> > >         Args:
> > >             erf_map: numpy array, shape (H, W)
> > >             filename: path to output image, e.g., "erf_hot.png"
> > >             colormap: color mapping, e.g., 'gray', 'hot', 'jet', 'viridis', 'plasma'
> > >         """
> > >         import matplotlib.pyplot as plt
> > >
> > >         plt.figure(figsize=(6, 6))
> > >         plt.imshow(erf_map, cmap=colormap)
> > >         plt.axis('off')  # Remove axes
> > >         plt.colorbar(fraction=0.046, pad=0.04)
> > >         plt.tight_layout()
> > >         plt.savefig(filename, dpi=200, bbox_inches='tight', pad_inches=0)
> > >         plt.close()
> > >         print(f"ERF map saved as '{filename}'")
> > >
> > >
> > > ---
> > >
> > >
> > > - **Numerical ERF Maps**
> > >     - Due to rebuttal space constraints and the original ERF map resolution of (224, 224), we apply spatial average pooling to reduce the shape to **(28, 28)** and round each value to **3 decimal** places. This allows us to present the full ERF maps as raw data in plain text.
> > >     - We will post the **pooled ERF values** for both ViM and AMViM models **in the following two comments.**
> > >
> > > We hope this workaround helps convey the intended visual information in a text-compliant way under current NeurIPS policies.

---

> > > ### Author Response · Authors · 2025-08-05
> > >
> > > **Pooled ERF Values (Vim)**
> > >
> > > The pooled ERF of the ViM model is provided below：
> > >
> > > ---
> > >
> > > [[0.198,0.188,0.046,0.023,0.033,0.029,0.007,0.007,0.012,0.012,0.019,0.024,0.013,0.012,0.008,0.011,0.019,0.020,0.025,0.026,0.012,0.008,0.018,0.021,0.009,0.011,0.015,0.017],
> > >  [0.168,0.240,0.042,0.019,0.031,0.033,0.007,0.006,0.012,0.009,0.017,0.021,0.015,0.012,0.006,0.008,0.015,0.019,0.018,0.019,0.011,0.010,0.029,0.020,0.007,0.008,0.012,0.019],
> > >  [0.062,0.048,0.018,0.009,0.007,0.006,0.011,0.009,0.012,0.011,0.004,0.007,0.012,0.012,0.005,0.005,0.009,0.010,0.007,0.007,0.003,0.004,0.005,0.005,0.003,0.004,0.015,0.012],
> > >  [0.046,0.050,0.025,0.010,0.008,0.008,0.009,0.008,0.014,0.009,0.005,0.006,0.009,0.012,0.006,0.005,0.009,0.011,0.007,0.006,0.003,0.003,0.004,0.004,0.003,0.004,0.018,0.017],
> > >  [0.124,0.098,0.019,0.016,0.015,0.010,0.004,0.004,0.004,0.006,0.005,0.006,0.005,0.005,0.004,0.004,0.004,0.005,0.012,0.013,0.014,0.015,0.006,0.006,0.012,0.015,0.018,0.016],
> > >  [0.102,0.095,0.019,0.019,0.009,0.008,0.003,0.004,0.005,0.006,0.007,0.005,0.005,0.004,0.003,0.004,0.005,0.004,0.011,0.014,0.014,0.014,0.006,0.005,0.011,0.013,0.016,0.016],
> > >  [0.110,0.137,0.048,0.043,0.017,0.014,0.008,0.008,0.009,0.009,0.004,0.004,0.004,0.003,0.002,0.002,0.008,0.008,0.011,0.010,0.006,0.005,0.005,0.005,0.003,0.005,0.012,0.012],
> > >  [0.135,0.139,0.047,0.051,0.019,0.019,0.009,0.008,0.008,0.007,0.006,0.005,0.003,0.004,0.002,0.002,0.007,0.008,0.009,0.009,0.006,0.004,0.005,0.005,0.003,0.004,0.017,0.015],
> > >  [0.040,0.035,0.018,0.023,0.013,0.010,0.005,0.005,0.003,0.003,0.003,0.003,0.008,0.009,0.004,0.003,0.004,0.003,0.004,0.004,0.004,0.003,0.003,0.004,0.003,0.004,0.005,0.007],
> > >  [0.031,0.036,0.022,0.024,0.008,0.009,0.005,0.005,0.003,0.003,0.003,0.004,0.007,0.007,0.004,0.004,0.004,0.004,0.003,0.003,0.005,0.005,0.003,0.004,0.003,0.003,0.006,0.006],
> > >  [0.010,0.010,0.007,0.005,0.004,0.003,0.002,0.002,0.002,0.002,0.003,0.004,0.008,0.008,0.007,0.008,0.005,0.005,0.006,0.006,0.031,0.024,0.009,0.008,0.009,0.010,0.009,0.008],
> > >  [0.011,0.008,0.008,0.006,0.003,0.004,0.002,0.002,0.002,0.002,0.003,0.004,0.009,0.009,0.009,0.008,0.005,0.005,0.005,0.006,0.022,0.023,0.012,0.009,0.009,0.008,0.007,0.009],
> > >  [0.023,0.024,0.023,0.020,0.011,0.010,0.006,0.005,0.003,0.003,0.003,0.004,0.008,0.010,0.008,0.005,0.004,0.004,0.004,0.004,0.005,0.005,0.003,0.003,0.006,0.007,0.011,0.014],
> > >  [0.020,0.019,0.021,0.019,0.014,0.010,0.006,0.006,0.003,0.002,0.003,0.005,0.010,0.008,0.007,0.006,0.003,0.004,0.004,0.005,0.005,0.006,0.003,0.005,0.006,0.007,0.011,0.014],
> > >  [0.007,0.007,0.011,0.008,0.008,0.008,0.006,0.007,0.006,0.005,0.005,0.004,0.006,0.006,0.006,0.006,0.003,0.003,0.003,0.003,0.005,0.004,0.004,0.004,0.004,0.005,0.013,0.014],
> > >  [0.008,0.008,0.009,0.006,0.006,0.007,0.006,0.007,0.006,0.005,0.004,0.006,0.006,0.006,0.005,0.004,0.004,0.002,0.003,0.004,0.004,0.003,0.004,0.004,0.004,0.005,0.013,0.014],
> > >  [0.005,0.006,0.006,0.006,0.005,0.007,0.008,0.008,0.005,0.008,0.003,0.004,0.004,0.004,0.005,0.003,0.004,0.004,0.006,0.006,0.003,0.003,0.003,0.002,0.006,0.005,0.014,0.016],
> > >  [0.005,0.006,0.006,0.006,0.005,0.006,0.010,0.007,0.007,0.005,0.003,0.003,0.004,0.004,0.005,0.004,0.004,0.004,0.005,0.006,0.003,0.003,0.002,0.002,0.006,0.005,0.013,0.020],
> > >  [0.005,0.005,0.013,0.009,0.023,0.022,0.015,0.013,0.005,0.005,0.005,0.006,0.005,0.006,0.009,0.008,0.004,0.003,0.004,0.003,0.003,0.004,0.004,0.003,0.008,0.011,0.016,0.019],
> > >  [0.004,0.005,0.011,0.009,0.030,0.026,0.016,0.012,0.006,0.005,0.006,0.005,0.004,0.005,0.010,0.008,0.004,0.005,0.005,0.005,0.003,0.003,0.005,0.005,0.010,0.013,0.021,0.025],
> > >  [0.021,0.021,0.018,0.014,0.007,0.007,0.006,0.005,0.007,0.005,0.006,0.006,0.007,0.006,0.006,0.005,0.009,0.010,0.007,0.006,0.007,0.007,0.006,0.006,0.014,0.019,0.031,0.039],
> > >  [0.017,0.028,0.018,0.015,0.006,0.007,0.004,0.005,0.007,0.006,0.009,0.007,0.006,0.005,0.004,0.005,0.012,0.012,0.007,0.006,0.008,0.007,0.005,0.007,0.015,0.017,0.032,0.042],
> > >  [0.021,0.021,0.008,0.007,0.011,0.008,0.017,0.013,0.007,0.007,0.020,0.018,0.037,0.029,0.013,0.010,0.013,0.015,0.012,0.015,0.005,0.005,0.011,0.012,0.011,0.010,0.016,0.014],
> > >  [0.017,0.022,0.007,0.006,0.009,0.009,0.016,0.015,0.008,0.008,0.020,0.022,0.037,0.039,0.016,0.011,0.013,0.010,0.009,0.010,0.005,0.005,0.012,0.012,0.013,0.012,0.018,0.023],
> > >  [0.020,0.019,0.009,0.007,0.008,0.007,0.011,0.010,0.007,0.005,0.016,0.017,0.023,0.030,0.017,0.011,0.009,0.010,0.014,0.016,0.032,0.031,0.013,0.014,0.023,0.025,0.056,0.067],
> > >  [0.025,0.020,0.012,0.010,0.007,0.009,0.013,0.009,0.006,0.006,0.019,0.015,0.027,0.029,0.014,0.013,0.010,0.006,0.013,0.016,0.035,0.024,0.013,0.017,0.032,0.029,0.084,0.067],
> > >  [0.193,0.150,0.051,0.063,0.026,0.027,0.056,0.053,0.017,0.018,0.009,0.012,0.025,0.021,0.014,0.013,0.008,0.010,0.018,0.019,0.033,0.047,0.040,0.047,0.085,0.096,0.091,0.112],
> > >  [0.152,0.183,0.058,0.065,0.028,0.041,0.067,0.065,0.025,0.017,0.009,0.013,0.026,0.022,0.013,0.012,0.011,0.009,0.018,0.017,0.050,0.044,0.041,0.050,0.142,0.145,0.118,0.164]]
> > >
> > > ---

---

> > > ### Author Response · Authors · 2025-08-05
> > >
> > > **Pooled ERF Values (AMVim)**
> > >
> > > The pooled ERF of our AMViM model is provided below：
> > >
> > > ---
> > >
> > > [[0.218,0.204,0.036,0.028,0.017,0.021,0.009,0.011,0.029,0.019,0.017,0.018,0.031,0.024,0.023,0.026,0.080,0.070,0.030,0.031,0.010,0.010,0.011,0.016,0.011,0.021,0.022,0.030],
> > >  [0.189,0.212,0.036,0.027,0.016,0.017,0.009,0.011,0.028,0.029,0.013,0.016,0.028,0.030,0.015,0.016,0.049,0.039,0.025,0.030,0.012,0.012,0.014,0.013,0.010,0.016,0.025,0.035],
> > >  [0.070,0.061,0.028,0.021,0.026,0.016,0.007,0.006,0.012,0.011,0.005,0.005,0.030,0.027,0.007,0.007,0.013,0.019,0.018,0.015,0.006,0.006,0.007,0.006,0.010,0.011,0.051,0.056],
> > >  [0.070,0.058,0.024,0.025,0.024,0.022,0.007,0.007,0.012,0.009,0.007,0.008,0.018,0.021,0.011,0.007,0.013,0.016,0.017,0.014,0.006,0.005,0.005,0.007,0.008,0.008,0.053,0.072],
> > >  [0.124,0.099,0.026,0.027,0.032,0.025,0.007,0.006,0.005,0.006,0.008,0.008,0.004,0.006,0.005,0.006,0.008,0.010,0.028,0.028,0.040,0.030,0.012,0.010,0.017,0.020,0.037,0.039],
> > >  [0.111,0.087,0.025,0.031,0.023,0.019,0.007,0.007,0.006,0.006,0.008,0.007,0.005,0.005,0.005,0.006,0.008,0.008,0.023,0.024,0.026,0.030,0.014,0.009,0.016,0.016,0.030,0.037],
> > >  [0.201,0.200,0.039,0.034,0.016,0.011,0.006,0.006,0.008,0.009,0.006,0.005,0.004,0.003,0.003,0.003,0.013,0.015,0.024,0.015,0.008,0.006,0.008,0.010,0.007,0.008,0.038,0.046],
> > >  [0.243,0.209,0.050,0.039,0.018,0.015,0.006,0.008,0.008,0.006,0.005,0.005,0.004,0.003,0.005,0.004,0.015,0.013,0.021,0.018,0.006,0.006,0.010,0.011,0.007,0.008,0.029,0.035],
> > >  [0.044,0.043,0.015,0.019,0.010,0.012,0.005,0.005,0.007,0.007,0.006,0.006,0.008,0.006,0.006,0.006,0.008,0.006,0.013,0.009,0.005,0.004,0.006,0.007,0.006,0.008,0.016,0.019],
> > >  [0.041,0.048,0.018,0.021,0.011,0.011,0.005,0.006,0.008,0.008,0.005,0.005,0.007,0.007,0.006,0.006,0.008,0.008,0.012,0.012,0.005,0.004,0.007,0.008,0.006,0.007,0.017,0.020],
> > >  [0.016,0.016,0.012,0.016,0.012,0.010,0.006,0.005,0.004,0.004,0.004,0.003,0.015,0.014,0.008,0.008,0.006,0.007,0.006,0.005,0.011,0.007,0.005,0.006,0.008,0.008,0.010,0.016],
> > >  [0.014,0.013,0.014,0.015,0.012,0.010,0.005,0.006,0.005,0.005,0.003,0.004,0.016,0.013,0.008,0.009,0.007,0.008,0.005,0.006,0.007,0.008,0.006,0.006,0.008,0.009,0.012,0.013],
> > >  [0.035,0.031,0.032,0.028,0.021,0.019,0.012,0.018,0.004,0.004,0.005,0.008,0.013,0.010,0.024,0.021,0.008,0.008,0.007,0.007,0.006,0.006,0.008,0.007,0.013,0.009,0.020,0.017],
> > >  [0.028,0.026,0.029,0.026,0.022,0.019,0.011,0.013,0.005,0.004,0.005,0.005,0.010,0.012,0.024,0.025,0.007,0.009,0.006,0.006,0.005,0.007,0.006,0.008,0.013,0.011,0.018,0.016],
> > >  [0.018,0.016,0.025,0.025,0.013,0.010,0.008,0.007,0.005,0.006,0.015,0.011,0.006,0.005,0.006,0.006,0.004,0.005,0.005,0.006,0.008,0.007,0.010,0.013,0.017,0.017,0.015,0.015],
> > >  [0.016,0.017,0.027,0.023,0.010,0.009,0.006,0.006,0.006,0.005,0.014,0.016,0.005,0.005,0.007,0.005,0.004,0.005,0.006,0.007,0.007,0.006,0.011,0.011,0.011,0.013,0.013,0.016],
> > >  [0.015,0.015,0.007,0.008,0.008,0.008,0.012,0.010,0.024,0.023,0.015,0.013,0.005,0.004,0.007,0.006,0.004,0.004,0.004,0.004,0.003,0.003,0.005,0.005,0.013,0.012,0.030,0.036],
> > >  [0.017,0.017,0.007,0.008,0.008,0.009,0.011,0.012,0.029,0.025,0.014,0.015,0.005,0.005,0.007,0.006,0.005,0.005,0.004,0.005,0.003,0.003,0.005,0.006,0.011,0.013,0.029,0.044],
> > >  [0.012,0.012,0.009,0.010,0.011,0.010,0.011,0.012,0.007,0.006,0.008,0.009,0.005,0.006,0.009,0.010,0.006,0.007,0.005,0.005,0.004,0.006,0.011,0.009,0.018,0.024,0.036,0.049],
> > >  [0.013,0.012,0.010,0.011,0.015,0.011,0.011,0.010,0.006,0.006,0.010,0.010,0.005,0.005,0.006,0.008,0.006,0.007,0.006,0.006,0.005,0.005,0.009,0.009,0.026,0.032,0.040,0.059],
> > >  [0.019,0.018,0.015,0.010,0.009,0.008,0.007,0.011,0.009,0.005,0.004,0.004,0.006,0.006,0.005,0.006,0.003,0.003,0.006,0.005,0.006,0.005,0.009,0.010,0.025,0.031,0.090,0.103],
> > >  [0.014,0.016,0.010,0.013,0.008,0.008,0.009,0.011,0.008,0.008,0.007,0.004,0.005,0.006,0.005,0.005,0.004,0.004,0.005,0.006,0.007,0.005,0.009,0.010,0.018,0.025,0.091,0.094],
> > >  [0.060,0.038,0.011,0.011,0.011,0.009,0.027,0.030,0.013,0.014,0.019,0.020,0.015,0.012,0.005,0.007,0.005,0.005,0.010,0.012,0.032,0.022,0.026,0.022,0.026,0.028,0.041,0.052],
> > >  [0.039,0.035,0.011,0.015,0.016,0.009,0.027,0.030,0.011,0.012,0.026,0.021,0.014,0.012,0.006,0.007,0.005,0.006,0.010,0.011,0.030,0.023,0.023,0.026,0.026,0.030,0.070,0.100],
> > >  [0.027,0.019,0.015,0.015,0.014,0.013,0.015,0.017,0.008,0.008,0.012,0.014,0.021,0.025,0.015,0.011,0.017,0.014,0.027,0.032,0.079,0.082,0.027,0.027,0.042,0.035,0.069,0.106],
> > >  [0.030,0.022,0.016,0.018,0.016,0.011,0.020,0.017,0.009,0.009,0.013,0.011,0.031,0.034,0.012,0.012,0.016,0.017,0.031,0.030,0.091,0.074,0.025,0.023,0.043,0.046,0.113,0.119],
> > >  [0.089,0.067,0.045,0.039,0.060,0.068,0.071,0.065,0.017,0.016,0.008,0.009,0.027,0.025,0.011,0.009,0.017,0.023,0.078,0.072,0.080,0.087,0.059,0.060,0.124,0.139,0.138,0.169],
> > >  [0.105,0.135,0.061,0.062,0.066,0.082,0.068,0.074,0.015,0.018,0.008,0.010,0.027,0.024,0.012,0.013,0.019,0.029,0.103,0.088,0.095,0.087,0.061,0.059,0.221,0.250,0.159,0.247]]
> > >
> > > ---

---

### Official Review · Reviewer_S9i4 · 2025-07-02

**Clarity:** 3
**Significance:** 3
**Originality:** 3
**Rating:** 4
**Confidence:** 5

**Summary:**

This paper proposes a new Mamba-based vision model, Asymmetric Multi-scale Vision Mamba (AMVim), to address the accuracy degradation that occurs when applying token reduction techniques to previous Mamba-based vision models. There are two core contributions in AMVim: asymmetric scanning paths to effectively mitigate the drop in token connection survival rate when pruning or merging tokens, and a window-aware scanning mechanism that helps capture local information. In this way, there are two different paths in the proposed Window-based Multi-Scale State Space (WMS3) block, one for global context and another for local dependencies. Experiments on ImageNet-1K demonstrate that AMVim achieves state-of-the-art pruning robustness, with AMVim-T showing a 34% improvement in training-free accuracy and AMVim-S experiencing only a 1.5% drop in accuracy during token reduction, outperforming Vim and even matching ViT in some cases, while also excelling in pruning-free settings.

**Questions:**

- As for the window-based multi-scale selective scanning mechanism, there is no overlap between different windows. Will it be helpful to add some new modules to exchange the information between different windows? Just like the shift in the swin transformer.

**Ethical Concerns:**

["NO or VERY MINOR ethics concerns only"]

**Final Justification:**

From the throughput-vs-accuracy results, there is a big drop in accuracy (6%) with only a 1.1x speedup. So I will keep my rating.

**Limitations:**

yes

**Paper Formatting Concerns:**

No paper formatting concern.

**Quality:**

3

**Strengths And Weaknesses:**

- Strengths
	- This paper is well-written and easy to follow.
	- Extensive evaluations make the proposed method solid and help readers understand the model architecture design. And the extensive ablation studies on all relevant components make it clear.
- Weakness
	- This paper only evaluated the new model from the perspective of model accuracy, but it should also be important to test its real performance and speedup, not just by using FLOPs. And it'd be better to analyze the performance overhead of the proposed WMS3 block.
	- The evaluations are only conducted on the image classification task. It would be better to show more results on other benchmarks.

---

> ### Author Rebuttal · Authors · 2025-07-30
>
> We sincerely thank Reviewer S9i4 for the insightful suggestions and constructive feedback.
> > **Q1**: This paper only evaluated the new model from the perspective of model accuracy, but it should also be important to test its real performance and speedup, not just by using FLOPs. And it'd be better to analyze the performance overhead of the proposed WMS3 block
>
> **A1**: Thanks for the valuable suggestion. In response, we provide a detailed analysis of real hardware performance by reporting throughput-vs-accuracy results.
>
> **Setting**: All experiments are conducted fairly on a single NVIDIA GeForce RTX 4090 GPU with an image resolution of 224×224. We use ToMe as the default token reduction strategy. We choose Vim and LocalVim (window-scan based) as baselines.
>
> **Throughout (image/s):**
>
>
> | method \ reduction ratio |   0.00    | 0.17   | 0.27   | 0.36   | 0.46   |
> |:----------------------:|:------:| :------: | :------: | :------: | :------: |
> |          Vim           | 2685.2 | 2630.9 | 2962.0 | 3291.0 | 3416.7 |
> |        LocalVim        | 955.3  | 1065.5 | 1168.0 | 1220.2 | 1548.5 |
> |         AMVim          | 2497.3 | 2452.5 | 2750.7 | 3063.1 | 3278.6 |
>
>
> **Top-1 Accuracy (%):**
> | method \ reduction ratio |   0.00    | 0.17   | 0.27   | 0.36   | 0.46   |
> |:----------------------:|:------:| :------: | :------: | :------: | :------: |
> |          Vim           | 76.1| 38.7 | 41.6 | 37.6 | 34.2 |
> |        LocalVim        | 76.2  | 60.4 | 50.9 | 43.3 | 36.2 |
> |         AMVim          | **76.3** | **72.8** | **70.3** | **66.7** |**61.3** |
>
> **Analysis:**
> - **Efficiency:** AMVim incurs only a modest throughput drop (~180 img/s on average) compared to Vim. This slight overhead mainly stems from the directional switching introduced in one branch of the WMS3 block. However, AMVim achieves **approximately 2.5× higher throughput** than LocalVim, which applies per-layer scanning configurations that introduce greater computational overhead.
> - **Robustness:** Across all token reduction ratios, AMVim consistently achieves the highest accuracy, significantly outperforming both Vim and LocalVim. This confirms the robustness of the proposed architecture under aggressive token pruning.
> - **Conclusion:** These results demonstrate that **AMVim strikes a significantly better balance between efficiency and pruning robustness**, making it a more practical and deployable solution for real-world applications.
>
>
>
> ---
>
>
> >**Q2:** The evaluations are only conducted on the image classification task. It would be better to show more results on other benchmarks.
>
> **A2:** Thanks for the helpful suggestion. In response, we add semantic segmentation as a downstream task to further validate the generalization ability of our method beyond image classification.
>
> **Semantic segmentation：**
>
> **Setting:** As no downstream-task checkpoints of Vim are publicly available, we follow the official codebase settings to fairly evaluate both Vim and AMVim. ToMe is used as the token reduction strategy, with training-free pruning results reported under identical conditions.
>
> **results:**
> | method  |   backbone    | image size | param (M) | val mIoU (%) |    $\triangle$     |
> |:-------:|:-------------:|:----------:|:-----:|:--------:|:------------------:|
> | UperNet |    Vim-Ti     |  $512^2$   |  13M  |   40.0   |         0          |
> | UperNet |     Vim-S     |  $512^2$   |  46M  |   43.3   |         0          |
> | UperNet |   AMVim-Ti    |  $512^2$   |  13M  | **40.2** | $\uparrow$ **0.2** |
> | UperNet |    AMVim-S    |  $512^2$   |  46M  | **43.4** | $\uparrow$ **0.1** |
> | UperNet |  ToMe-Vim-Ti  |  $512^2$   |  13M  |   21.1   |         0          |
> | UperNet |  ToMe-Vim-S   |  $512^2$   |  46M  |   22.0   |         0          |
> | UperNet | ToMe-AMVim-Ti |  $512^2$   |  13M  | **27.0** | $\uparrow$ **5.9** |
> | UperNet | ToMe-AMVim-S  |  $512^2$   |  46M  | **28.7** | $\uparrow$ **6.7** |
>
> **Analysis:**
>   - AMVim achieves better performance than Vim in the non-pruned setting, indicating stronger representational capacity.
>   - Under token reduction, AMVim consistently outperforms Vim by a significant margin (**up to +6.7 mIoU**), validating its robustness and generalization ability in dense prediction tasks.
>
> **Object detection：**
> Given the lack of publicly available Vim checkpoints for object detection, we would need to re-train both Vim and AMVim from scratch. Given the high resolution (1024×1024), each run takes **~8 days on 4× RTX 4090 GPUs.**
>
> Due to limited computational resources and other experiment priorities, this evaluation could not be completed within the rebuttal period. However, we acknowledge its importance and **plan to include it in the final version**.
>
>
>
> ---
> >**Q3:** As for the window-based multi-scale selective scanning mechanism, there is no overlap between different windows. Will it be helpful to add some new modules to exchange the information between different windows? Just like the shift in the swin transformer.
>
> **A3:** Thanks for the insightful suggestion. To investigate the potential benefits of cross-window information exchange, we incorporated a Swin-style shift module into our architecture.
>
> **Setting:**
> The shift is applied to the window-based branch. Specifically, odd-numbered WMS3 blocks perform cyclic shifts to enable cross-window interaction, while even-numbered blocks retain standard window-based scanning. We fine-tuned this modified variant from our AMVim checkpoint using the same training setup. ToMe is adopted as the pruning strategy, and we report training-free results.
>
> **Results:**
>
> | method            | image size | param (M) | FLOPs (G) | top-1 acc. (%) | $\triangle$      |
> |:----------------- |:----------:|:---------:| --------- |:--------------:|:---------------- |
> | AMVim-T           |  $224^2$   |     7     | 1.5       |    **76.3**    | 0                |
> | AMVim-T-swin      |  $224^2$   |     7     | 1.5       |      73.7      | $\downarrow$ 2.6 |
> | ToMe-AMVim-T      |  $224^2$   |     7     | 1.3       |   **72.8**    | 0                |
> | ToMe-AMVim-T-swin |  $224^2$   |     7     | 1.3       |      66.6      | $\downarrow$ 6.2 |
>
> **Analysis:**
> - Integrating the Swin-style shift results in a clear drop in both accuracy (↓2.6%) and pruning robustness (↓6.2%). This may stem from a misalignment between the uniform cyclic shift and the spatially structured design of our multi-scale mechanism, potentially disrupting intended token interactions.
> - We acknowledge the importance of cross-window communication. However, directly transferring the Swin shift from Swin Transformer is suboptimal in our setting. We will investigate more tailored and compatible strategies in future work.
>
> ---
> Thanks again for the professional, detailed, and valuable reviews! We have done our best to address each of the above concerns and hope our response can resolve them. Please let us know if there are any other questions. We will actively join the discussion until the end of the rebuttal period.

---

> > ### Author Response · Authors · 2025-08-05
> >
> > Dear Reviewer S9i4,
> >
> > We hope our rebuttal addresses all your concerns. To save your time, we summarize our key responses below:
> > - **Real-World Latency**
> >     -  We conducted detailed throughput-vs-accuracy evaluations on an NVIDIA RTX 4090 GPU.
> >     - AMVim strikes **a superior balance between robustness and efficiency**, confirming practical deployability.
> >
> > - **Downstream Generalization**
> >     - We added semantic segmentation experiments to test generalization beyond classification.
> >     - AMVim shows consistent gains under pruning, with **up to +6.7 mIoU**, validating robustness on dense prediction tasks.
> >
> > - **Cross-Window Information Exchange**
> >     - We implemented a Swin-style shift within the WMS3 module to enable cross-window communication.
> >     - This design led to a noticeable drop in accuracy, indicating that a direct transfer of Swin’s shifting mechanism is suboptimal for our architecture. We will explore more compatible alternatives in future work.
> >
> > Thank you again for your thoughtful and constructive feedback. We would be happy to discuss further during the rebuttal phase.

---

> > ### Comment · Reviewer_S9i4 · 2025-08-06
> >
> > Thanks for the detailed feedback. Most of my concerns are addressed. But from the throughput-vs-accuracy results, there is a big drop in accuracy (6%) with only a 1.1x speedup. So I will keep my rating.

---

### Official Review · Reviewer_FyAa · 2025-07-03

**Clarity:** 3
**Significance:** 3
**Originality:** 3
**Rating:** 4
**Confidence:** 4

**Summary:**

This paper proposes a novel architecture as a variant of Vim, named Asymmetric Multi-scale Vision Mamba (AMVim). The core insight is that the proposed asymmetric scanning mechanism outperforms the symmetric scanning strategy used in Vim. In addition, a window-aware scanning mechanism is introduced to further enhance model performance. The proposed method is evaluated on the ImageNet-1K dataset, along with extensive ablation studies.

**Questions:**

As outlined in the Weaknesses section above.

**Ethical Concerns:**

["NO or VERY MINOR ethics concerns only"]

**Final Justification:**

All my concerns have been resolved. Therefore, I have decided to maintain my decision of borderline accept.

**Limitations:**

Yes.

**Paper Formatting Concerns:**

None.

**Quality:**

3

**Strengths And Weaknesses:**

**Strengths**
1. The paper is well-organized and clearly written.
2. It proposes a novel pruning-robust architecture, AMVim, featuring an innovative asymmetric and window-aware scanning mechanism.
3. Comprehensive ablation studies are conducted to demonstrate the effectiveness of each component.
---
**Weaknesses**
1. **Inappropriate comparison setting in experiments.** In Table 1, it is not fair to compare the performance of AMVim with token-pruned results of Vim. A more appropriate comparison would involve evaluating AMVim against various Vim variants without pruning to establish its baseline performance. For pruning robustness, it would be more convincing to compare both Vim and AMVim under the same pruning strategies—such as Token Recognition and Hidden State Alignment. Including such results would strengthen the empirical support.
2. **Inconsistency in experimental results.** In Table 1, the top-1 accuracy of AMViM-Tiny is reported as 75.3, while in Table 3, the same model is shown to achieve 76.3. Since these refer to the same architecture, this discrepancy should be clarified.

---

> ### Author Rebuttal · Authors · 2025-07-29
>
> We sincerely thank Reviewer FyAa for the valuable comments and suggestions regarding the presentation of Table 1. Below, we provide our detailed response and corresponding modifications.
> > **Q1:** Inappropriate comparison setting in experiments. In Table 1, it is not fair to compare the performance of AMVim with token-pruned results of Vim. A more appropriate comparison would involve evaluating AMVim against various Vim variants without pruning to establish its baseline performance. For pruning robustness, it would be more convincing to compare both Vim and AMVim under the same pruning strategies—such as Token Recognition and Hidden State Alignment. Including such results would strengthen the empirical support.
>
> **A1:** Thanks for pointing out this confusion. We clarify the setting and motivation behind Table 1 as follows:
>
> **Setting:** The Top-1 accuracies reported in Table 1 correspond to AMVim with **ToMe** under a **reduction ratio of 0.17**, and are obtained after **fine-tuning**. These results are derived follow the Hidden State Alignment setting, as mentioned at the end of the first paragraph in Section 4.2. In contrast, all other results presented in the tables and figures (outside of Table 1) are based on training-free evaluations.
> To avoid confusion, we will revise Table 1 in the final version for greater clarity. The updated table is shown below:
>
>
> |        method         | reduction ratio | top1-acc. (%) | top1-acc. (%) | FLOPs (G) | FLOPs (G) |
> |:---------------------:|:---------------:|:-------------:|:-------------:|:---------:|:---------:|
> |                       |                 |   **tiny**    |   **tiny**    | **small** | **small** |
> |    Vim (baseline)     |      0.00       |     76.1      |     80.5      |   1.45    |   5.08    |
> |   Token Recognition   |      0.17       |     71.3      |     74.8      |   1.28    |   3.57    |
> | Hidden State Alignmen |      0.17       |     75.1      |     78.8      |   1.29    |   3.60    |
> |  **AMVim-ToMe (ft)**  |    **0.17**     |   **75.3**    |   **79.5**    | **1.27**  | **3.60**  |
>
>
>
> **Analysis:**
> - Instead of relying on complex, task-specific pruning techniques, improving the architectural resilience of Mamba enables genric pruning strategies (e.g., ToMe, originally designed for ViTs) to achieve promising performance.
> - This finding suggests that resolving Mamba’s architectural fragility offers a more flexible and scalable solution for pruning robustness.
>
>
>
> ---
>
>
> >**Q2**: Inconsistency in experimental results. In Table 1, the top-1 accuracy of AMViM-Tiny is reported as 75.3, while in Table 3, the same model is shown to achieve 76.3. Since these refer to the same architecture, this discrepancy should be clarified.
>
>
> **A2**: Thanks for the question. As clarified in **A1**, the 75.3% top-1 accuracy reported in Table 1 refers to AMVim with **ToMe**, under a **token reduction ratio of 0.17**, and is obtained after **fine-tuning**.
>
> In contrast, Table 3 reports the 76.3% accuracy as the **non-pruned** (full-token) baseline performance of AMVim, without applying any token reduction. Additionally, the 72.8% result in Table 3 reflects the **training-free** accuracy of AMVim with ToMe under the same 0.17 reduction ratio.
>
> ---
> Thanks again for helping us improve the paper, and we hope the response can resolve these concerns! Please let us know if there are any further questions. We will be actively available until the end of the rebuttal period.

---

> > ### Author Response · Authors · 2025-08-05
> >
> > Dear Reviewer FyAa,
> >
> > We hope our rebuttal addresses all your concerns. To save your time, we summarize our key responses below:
> > - **Comparison Setting in Table 1**
> >     -  We clarified that Table 1 presents **fine-tuned** AMVim results using ToMe (reduction ratio 0.17) under the Hidden State Alignment setting, whereas all other results are from **training-free** evaluations.
> >     - The table has been revised for clarity, and the updated version is included in our rebuttal for your reference.
> >
> > - **Motivation behind Table 1**
> >     - Our goal is to show that enhancing Mamba's architectural robustness allows even general pruning strategies like ToMe (designed for ViTs) to perform competitively.
> >     - This suggests that solving fragility at the architecture level can be more flexible and scalable than designing mamba-specific pruning methods.
> >
> > Thank you again for your thoughtful and constructive feedback. We would be happy to discuss further during the rebuttal phase.

---

> > ### Comment · Reviewer_FyAa · 2025-08-05
> >
> > I appreciate the authors' detailed and thoughtful responses. Most of my concerns have been adequately addressed. Overall, I will maintain my original rating and wish the authors the best with their submission.

---

### Note · Authors · 2025-08-14

We thank all reviewers for their thoughtful and constructive feedback. We are pleased that our work was recognized as well-written (Reviewers FyAa, S9i4, tmKq), novel (Reviewer FyAa), easy to implement (Reviewer tmKq), and supported by solid experimental validation (Reviewers FyAa, S9i4, f6Uw, tmKq). We also appreciate that the reviewers acknowledged our rebuttal addressed all raised concerns.

Regarding Reviewer f6Uw’s suggestion to upload supplementary PDFs, we regret that NeurIPS 2025 policy does not permit including supplementary PDFs, images, or rich media in the rebuttal. To support our claims, we have provided the numerical values for the pooled Effective Receptive Field (ERF) maps of Vim and AMVim in the comments, along with the visualization code for independent reproduction.

With respect to Reviewer S9i4’s note on the accuracy drop at 1.1× speedup, we clarify that under identical conditions, the baseline Vim model suffers a 34.5% drop, whereas AMVim only loses ~6%. This substantial gap demonstrates that AMVim delivers efficiency gains with markedly smaller performance loss, highlighting its strong pruning resilience.

We thank the reviewers and the area chair once again for their time and insightful comments, which have greatly strengthened our work. We look forward to the opportunity to share AMVim with the NeurIPS community.

---

### Decision · Program_Chairs · 2025-09-17

**Decision:**

Accept (poster)

**Comment:**

This paper received 4, 3, 4, 4 originally, which are elevated to 4, 5, 5, 4 after rebuttal and discussions. The main contribution is to modify the vision Mamba such that it becomes robust to pruning. The reviewers all praised the well-written nature and comprehensive ablation studies. After the rebuttal, most concerns have been resolved, but there is still concerns that the throughput increase may not be worthy the reduction in accuracy. However, overall reviewers all agree that the current method is already a valuable contribution to the community.